# Protective role of renal proximal tubular alpha-synuclein in the pathogenesis of kidney fibrosis

Milica Bozic[1✉], Maite Caus [1], Raul R. Rodrigues-Diez [2], Neus Pedraza [3], Marta Ruiz-Ortega[2], Eloi Garí[3], Pilar Gallel[4], Maria José Panadés[4], Ana Martinez[1], Elvira Fernández[1] & José Manuel Valdivielso[1✉]

Kidney fibrosis is a highly deleterious process and a final manifestation of chronic kidney disease. Alpha-(α)-synuclein (SNCA) is an actin-binding neuronal protein with various functions within the brain; however, its role in other tissues is unknown. Here, we describe the expression of SNCA in renal epithelial cells and demonstrate its decrease in renal tubules of murine and human fibrotic kidneys, as well as its downregulation in renal proximal tubular epithelial cells (RPTECs) after TGF-β1 treatment. shRNA-mediated knockdown of SNCA in RPTECs results in de novo expression of vimentin and α-SMA, while SNCA overexpression represses TGF-β1-induced mesenchymal markers. Conditional gene silencing of SNCA in RPTECs leads to an exacerbated tubulointerstitial fibrosis (TIF) in two unrelated in vivo fibrotic models, which is associated with an increased activation of MAPK-p38 and PI3K-Akt pathways. Our study provides an evidence that disruption of SNCA signaling in RPTECs contributes to the pathogenesis of renal TIF by facilitating partial epithelial-to-mesenchymal transition and extracellular matrix accumulation.

[1] Vascular and Renal Translational Research Group, Institute for Biomedical Research in Lleida (IRBLleida) and RedInRen Retic, ISCIII, Spain. [2] Cellular and Molecular Biology in Renal and Vascular Pathology, IIS-Fundación Jiménez Díaz-Universidad Autónoma Madrid, Madrid, Spain. [3] Cell Cycle, Department of Basic Medical Science, IRBLleida, University of Lleida, Lleida, Spain. [4] Department of Pathology and Molecular Genetics, University Hospital Arnau de Vilanova and University of Lleida, IRBLleida, Spain. ✉email: milica.bozic@irblleida.udl.cat; josemanuel.valdivielso@udl.cat

F ibrotic disease is a dynamic and complex disorder char-
acterized by an excessive accumulation and deposition of
extracellular matrix, subsequently leading to a formation of a
fibrous scar, destruction of an organ parenchyma and the loss of
organ function[1,2]. There is currently no treatment for this devas-
tating condition. However, elucidation of the intricate cellular and
molecular pathways underlying organ fibrosis could lead to the
development of effective therapeutic strategies to cure, or at least
delay, organ deterioration[3]. One of the tissues in which fibrosis is
the final pathway to organ failure is the kidney. The pathogenesis
of renal fibrosis engages multiple molecular pathways and various
renal and infiltrating cell types[4]. Among many cellular protago-
nists of the renal fibrotic process, one of the significant roles was
assigned to renal proximal tubular epithelial cells (RPTECs)[3].
These cells are believed to be the major target for cellular injury
due to an exhaustive reabsorption of substantial volumes of fluid
and high enzymatic demands[5]. Being gifted by the unique
arrangements of the actin cytoskeleton[6], RPTECs respond
to injury by de novo expression of mesenchymal markers and
reorganization of the actin cytoskeleton[7,8], while still preserving
some of the epithelial characteristics[2,9], a process recently rede-
fined as partial epithelial-to-mesenchymal transition (EMT)[2,9].
Although still residing within the tubule, RPTECs through
paracrine signalling further promote inflammation, fibrosis, and
tissue damage[3].

Alpha (α)–synuclein (SNCA) is a member of the synuclein
family of structurally related proteins, composed of α-, β- and γ-
synuclein, and synoretin[10]. SNCA is preferentially expressed in
areas of the adult central nervous system (CNS) that exhibit
synaptic plasticity[11]. In addition to its distribution within the
CNS, it has been reported that SNCA is also expressed in a variety
of non-neuronal cells and tissues[12–18]. A comparative study of
SNCA expression in the developing and adult human peripheral
tissues reveals that once the specific organ development is fin-
ished, the levels of SNCA in most of the peripheral tissues
decrease, except in the kidney, adrenal gland, and testis, implying
that SNCA could perform important physiological function in
these organs[12].

Despite its implication in the pathogenesis of various synu-
cleinopathies[19], the exact physiological function of SNCA still
remains unclear. Several studies demonstrated the existence of a
neuroprotective function of native SNCA against oxidative
stress[20–22], neurodegeneration[23,24] and apoptosis[25,26]. SNCA has
been proposed to play a major role in actin cytoskeletal organi-
zation and modulation of microfilament function[27], which is
important for the normal functionality of neurons. Namely,
SNCA interacts with cytoskeletal components such as micro-
tubules[28] and actin[29] directly regulating function and dynamics of
actin cytoskeleton[27]. In rat hippocampus SNCA can activate the
N-methyl-D-aspartate receptor (NMDAR)[30], a protein assembly
that intimately associates with actin cytoskeleton[31]. Interestingly,
previous results of our group demonstrated that activation of
NMDAR in RPTECs decreased EMT in vitro and significantly
attenuated renal fibrosis in vivo, pointing to an indispensable role
of this receptor in the preservation of normal epithelial phenotype
of RPTECs[32,33].

In this report, we combine experimental studies using renal
tubular epithelial cells and conditional knockout mice with clin-
ical data from human renal tissues to assess and dissect the role of
renal tubular epithelial SNCA in kidney fibrosis. Here we show
that downregulation of SNCA enhances EMT in RPTECs in vitro
and that genetic ablation of epithelial SNCA results in an
aggravated fibrosis in two unrelated fibrosis models induced by
unilateral obstruction (UUO) and adenine rich diet. Our study
provides evidence for an important role of SNCA in preserving
the epithelial phenotype of RPTECs and in protecting kidney

parenchyma against injury. The results highlight the significance
of preserving the basal SNCA levels in the kidney as a therapeutic
strategy to attenuate the progression of kidney fibrosis.

## Results

**TGF-β1 treatment decreases SNCA expression in vitro**. We first
evaluated the expression levels of different members of the
synuclein family in HK-2 cells in basal conditions. We found
significantly higher expression of SNCA in HK-2 compared with
the levels of β-synuclein (SNCB) and γ-synuclein (SNCG) (Sup-
plementary Fig. 1A, B). We next assessed the expression of SNCA
in HK-2 cells upon treatment with TGF-β1, an important med-
iator of fibrosis signaling in renal epithelial cells[34,35]. Alongside
changes in the epithelial phenotype, evident as a loss of cobble-
stoned morphology (Supplementary Fig. 2A, B), decrease of E-
cadherin (Fig. 1b, c) and an increase of α-SMA and vimentin
(Fig. 1b and e–f), TGF-β1 induced a significant decrease of SNCA
mRNA (Fig. 1a; 48 and 72 h) and protein expression (Fig. 1b and
D; 72 h) in a dose- (Fig. 1a, d) and time-dependent manner
(Fig. 1a). The immunofluorescence showed a visible loss of SNCA
protein in the cytoplasm and nuclei of HK-2 cells treated with
TGF-β1 (72 h, Fig. 1g). These results suggest a dysregulation of
endogenous SNCA after TGF-β1 treatment in HK-2 and raise
the possibility of its potential role in maintaining the epithelial
phenotype of renal proximal tubular cells in vitro.

**Knockdown of SNCA influences RPTEC phenotype**. To con-
firm the possible role of basal SNCA levels in preserving the
epithelial phenotype in vitro, we first disrupted the expression of
SNCA by short hairpin RNA (shRNA). HK-2 were infected with
FSVsi-SNCA (shSNCA) or FSVsi (VC, control vector), and
subsequently incubated either with 2 ng/ml of TGF-β1 or were
left untreated for 72 h. HK-2 infected with FSVsi-SNCA showed
an evident decrease of SNCA expression compared with the
control (Fig. 2a, b), while mRNA and protein levels of SNCB and
SNCG did not change between VC and shSNCA (Supplementary
Fig. 1C–E). Interestingly, downregulation of SNCA in HK-2
(shSNCA) induced changes in the epithelial phenotype, evident as
a modest decrease of E-cadherin (Fig. 2a, c) and a marked
increase of α-SMA (Fig. 2a, e) and vimentin (Fig. 2a, d) compared
with the control (VC). Additionally, knockdown of SNCA
induced visible changes in actin organization seen as an increase
in Phalloidin fluorescence (Fig. 2k). Although TGF-β1 led to a
higher expression of mesenchymal markers (Fig. 2a and d–e) and
more pronounced reorganization of actin cytoskeleton (Fig. 2k)
in cells infected with control vector, this was not the case in
shSNCA cells, for which the levels of α-SMA and vimentin
expression were not further increased by TGF-β1 treatment and
were similar to those of VC + TGF-β1 condition (Fig. 2d, e).
These results reveal an important role of basal SNCA levels in
maintaining the epithelial phenotype of renal tubular cells.

**SNCA overexpression inhibits TGF-β1-induced α-SMA and
vimentin in vitro**. Next, we stably overexpressed SNCA in HK-2,
after which the cells were incubated with 2 ng/ml of TGF-β1 or
were left untreated for 72 h. Figure 2f and g shows an evident
increase of SNCA expression in HK-2 infected with SIN-PGK-
hsynuclein-WHV (SNCA Ox) compared with control vector
(VC), confirming the successful overexpression of the target
protein. TGF-β1 treatment induced a marked increase of α-SMA
and vimentin after 72 h of exposure (Fig. 2f, i–j), which was
significantly blunted in HK-2 cells overexpressing SNCA (two-
way ANOVA: $p < 0.05$). Overexpression of SNCA in HK-2 failed
to reestablish the normal expression of E-cadherin in cells treated
with TGF-β1 (Fig. 2f, h).

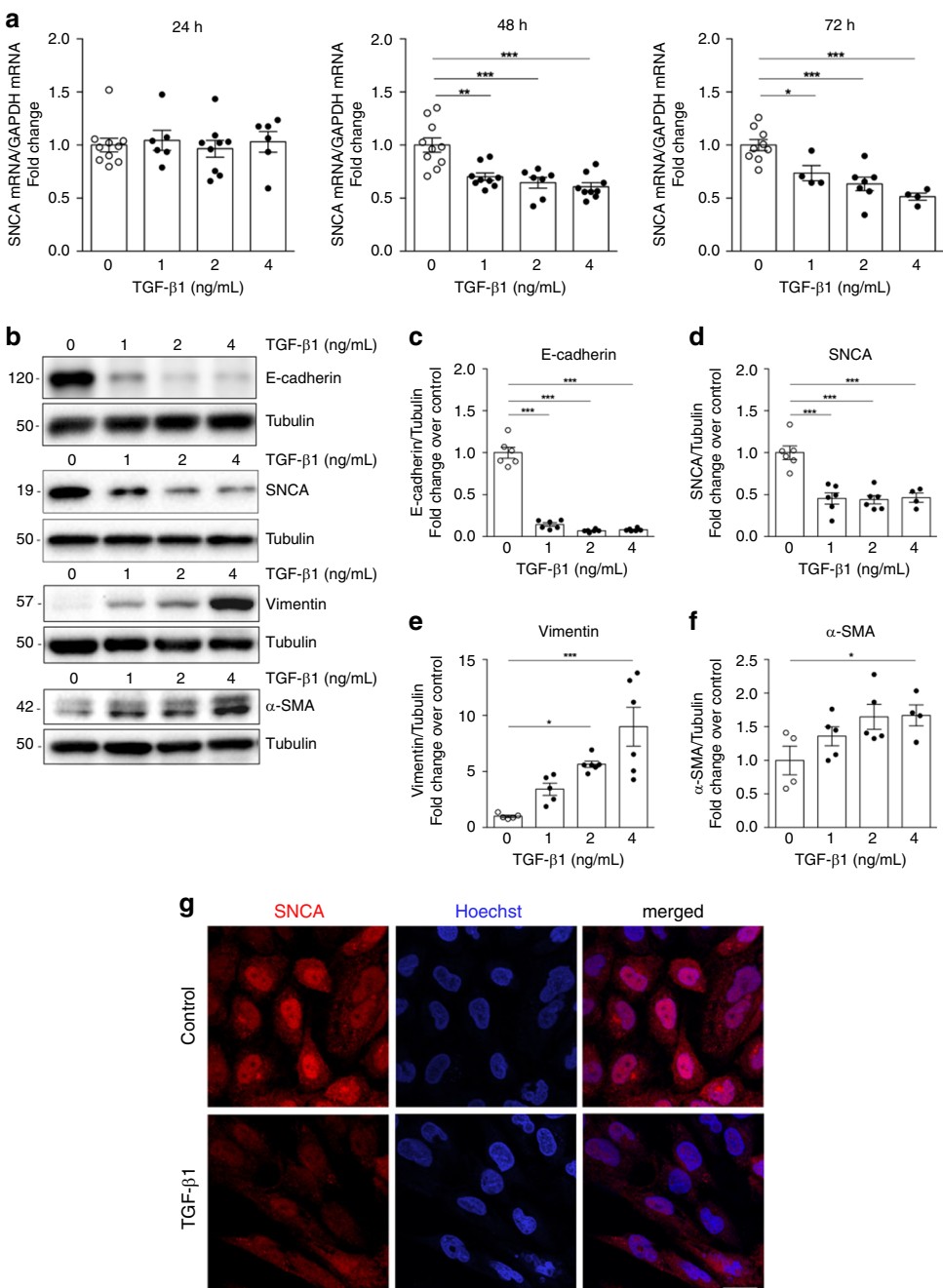

**Fig. 1 TGF-β1 leads to a decrease of SNCA expression in HK-2 in vitro.** Human renal proximal tubular epithelial (HK-2) cells were incubated in serum-free medium (control) or TGF-β1 for 24 h (**a**), 48 h (**a**) and 72 hours (**a, b–f**). **a** Total mRNA was extracted from HK-2 cells and mRNA levels of SNCA were determined by quantitative real-time PCR and normalized to GAPDH. Data are presented as mean ± SEM of at least $n = 3$ independent experiments. **b** Cell lysates were immunoblotted with antibodies against E-cadherin, SNCA, vimentin and α-SMA. The same samples were reprobed with antibodies against tubulin to ensure equal loading. Representative Western blots (**b**) and quantitative densitometric analysis (**c–f**) show decrease of E-cadherin and SNCA expression and an increase of vimentin and α-SMA expression in HK-2 cells treated with TGF-β1 for 72 h. Data are presented as mean ± SEM of at least $n = 2$ independent experiments. **g** Immunofluorescence staining for the distribution of SNCA in HK-2 cells after incubation with TGF-β1 for 72 h. Scale bar represents 20 μm. *$p < 0.05$, **$p < 0.01$, ***$p < 0.001$. The $p$-value by one-way ANOVA. Source data are provided as a Source Data file.

**SNCA modulates the activation of ERK1/2, Akt and p38 in vitro.** To study the potential cellular signaling pathway, we examined activities of MAPK and PI3K-Akt pathways, which are known to have critical role in the EMT of renal tubular cells and the progression of kidney fibrosis[32,36]. Knockdown of SNCA in HK-2 (shSNCA) induced an increase of pERK1/2, pAkt and p-p38 compared with control cells (VC), with pAkt and p-p38 reaching statistical significance (Fig. 3a, b). Although TGF-β1

treatment resulted in higher levels of pERK1/2, pAkt and p-p38 in cells infected with control vector, this was not the case in shSNCA cells, for which the levels of phosphorylated proteins were not further increased by the treatment (Fig. 3a, b). Overexpression of SNCA did not significantly affect the activity of pERK1/2, pAkt and p-p38 at basal state, nevertheless there was a tendency of decreasing the levels of p-p38 and pAkt in TGF-β1 treated cells (Supplementary Fig. 3A). To elucidate the possible mechanism

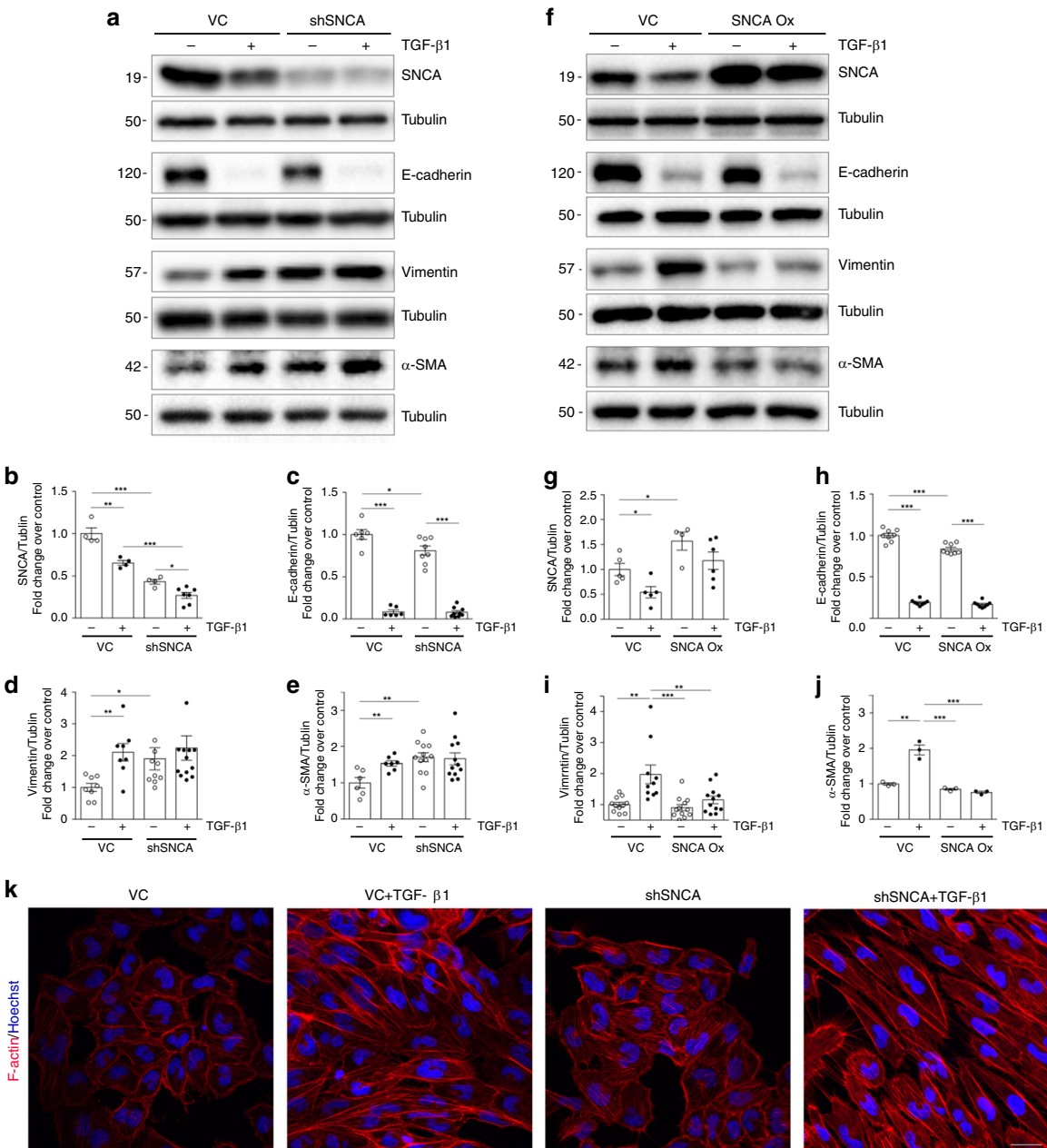

**Fig. 2 SNCA modulates the epithelial phenotype in vitro.** HK-2 cells (VC, shSNCA or SNCA Ox) were incubated with serum-free medium or treated with TGF-β1 (2 ng/ml) for 72 h. **a**, **f** Cell lysates were immunoblotted with antibodies against E-cadherin, SNCA, vimentin, and α-SMA. The same samples were reprobed with antibodies against tubulin to ensure equal loading. Representative Western blots (**a**, **f**) and quantitative densitometric analysis (**b–j**) show expression of SNCA, E-cadherin, α-SMA, and vimentin in VC and shSNCA cells (**a**, **b–e**) or VC and SNCA Ox cells (**f**, **g–j**) treated with TGF-β1 for 72 h. Data are presented as mean ± SEM of at least *n* = 3 independent experiments. **k** F-actin staining for the detection of actin filaments in HK-2 cells. Cells were incubated with serum-free medium or treated with TGF-β1 (2 ng/ml) for 24 h. F-actin was labeled with Phalloidin Alexa Fluor 568 (red). Nuclei were counterstained with Hoechst (blue). Scale bar represents 20 μm. *$p < 0.05$, **$p < 0.01$, ***$p < 0.001$. The p-value by two-way ANOVA. VC—control vector; shSNCA—cells with SNCA downregulation; SNCA Ox—cells overexpressing SNCA. Source data are provided as a Source Data file.

that stands behind the enhanced activity of MAPK pathway upon knockdown of SNCA, we first analyzed the expression of pMKK3 and pMKK6, two closely related dual-specificity protein kinases known to phosphorylate p38 MAP kinase. Knockdown of SNCA in HK-2 cells did not have any effect on the phosphorylation of MKK3/6, while the levels of p-p38 stayed increased (Supplementary Fig. 3B). To confirm that effects of SNCA knockdown on p38 MAPK phosphorylation are not due to MKK3/6 activity, we co-transfected cells with MKK6E and SNCA-flag plasmids. Co-transfection experiments showed that SNCA managed to

decrease the levels of phosphorylated p38 in cells with constitutive activation of MKK6 kinase (Supplementary Fig. 3 C, D), confirming that SNCA does not actuate through MKK3/6 kinase, but downstream of it. Our next step was to investigate the possible interaction between SNCA and p38 MAPK. As shown in Fig. 3f, p38 MAPK colocalized with SNCA in HK-2 cells in basal conditions. Moreover, co-immunoprecipitation experiments using cell lysates from HEK293T cells co-transfected with SNCA-flag and p38-HA confirmed the interaction of SNCA with p38 MAPK (Fig. 3g).

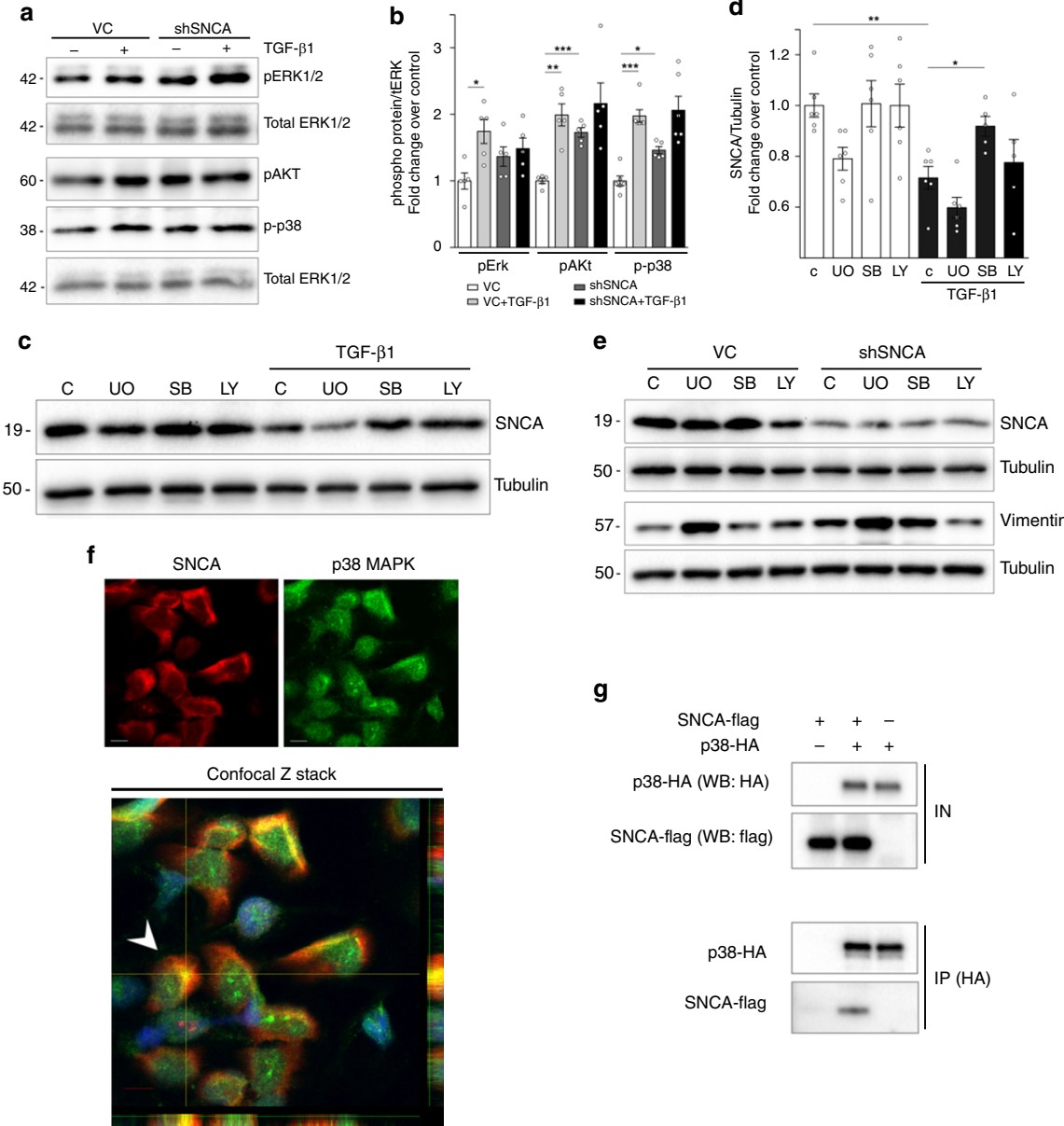

**Fig. 3 SNCA modulates the activation of ERK1/2, Akt and p38 in vitro. a, b** HK-2 cells were incubated separately with either serum-free medium or 2 ng/ml TGF-β1 for 60 min. Cell lysates were immunoblotted with antibodies against pERK1/2, p-p38, pAkt, total ERK1/2, and tubulin. Representative Western blots (**a**) and quantitative densitometric analysis (**b**) show levels of phosphorylated ERK1/2, p38, and Akt in HK-2. Data were normalized to total ERK1/2 (**b**) and presented as mean ± SEM of at least $n = 3$ independent experiments. **c**–**e** Effect of MAPK inhibitors (U0126 and SB203580) and PI3K-Akt inhibitor (LY294002) on TGF-β1-induced decrease in SNCA expression (**c, d**), as well as SNCA knockdown-induced increase of vimentin in HK-2 (**e**). HK-2 were pretreated for 1 h with UO126, (UO; 10 μM/L), SB203580 (SB; 20 μM/L) or LY294002 (LY; 10 μM/L), after which cells were incubated either with serum-free medium or TGF-β1 (2 ng/ml) for 72 h. Cell lysates were immunoblotted with antibodies against SNCA (**c, e**), vimentin (**e**) and tubulin. Representative Western blots (**c, e**) and quantitative densitometric analysis (**d**) show levels of SNCA (**c, e**) and vimentin (**e**) in HK-2. Data were normalized to tubulin (**d**) and presented as mean ± SEM of at least $n = 3$ independent experiments. **f** The confocal Z stack shows colocalization of p38 with SNCA in the cytoplasm of HK-2. HK-2 cells were grown under normal conditions and were stained using anti-SNCA (red) and anti-p38 antibody (green). Nuclei were labeled by Hoechst (blue). The lower and right panels in the confocal Z stack show a vertical cross section (yellow lines) through the cells. Arrow, SNCA colocalized with p38 MAPK in the cytoplasm. Scale bar represents 10 μm. **g** Immunoprecipitation of SNCA with p38. HEK293T cells were co-transfected with p38-HA and SNCA-flag. After crosslinking with DSP, p38 was immunoprecipitated from the cell lysates using antibodies against HA, as explained in Supplementary Methods. Lysates and immunoprecipitates were probed on Western blots with antibodies against flag and HA. *$p < 0.05$, **$p < 0.01$, ***$p < 0.001$. The $p$-value by two-way ANOVA. C—control. Source data are provided as a Source Data file.

Next, we sought to investigate the mechanism that stands behind the TGF-β1-induced decrease in SNCA expression. Treatment with p38 kinase inhibitor (SB203580) significantly attenuated TGF-β1-mediated decrease in SNCA expression (Fig. 3c, d). PI3K kinase inhibitor (LY294002) showed a tendency to decrease the TGF-β1-induced downregulation of SNCA (Fig. 3c, d), while MAPK-ERK1/2 kinase inhibitor (U0126) did not have any affect (Fig. 3c, d). These results suggest that MAPK-p38 signaling is one of the pathways implicated in the TGF-β1-mediated regulation of SNCA expression in HK-2 cells.

Next, we assessed the signaling pathway responsible for an increase in vimentin expression induced by a downregulation of SNCA in HK-2 cells. Interestingly, an increase of vimentin expression in shSNCA cells was blunted after treatment with PI3K kinase inhibitor (LY294002), while MAPK/ERK1/2 kinase inhibitor and p38 kinase inhibitor did not decrease its expression (Fig. 3e). These results point toward a role of PI3K-Akt pathway in the SNCA-mediated regulation of vimentin expression in HK-2 cells.

**SNCA expression decreases in obstructed mouse kidneys**. To investigate the potential role of SNCA in renal fibrosis, we subjected C57BL/6J wild-type mice to unilateral ureteral obstruction (UUO), a well-studied model of renal tubulointerstitial fibrosis and disease progression[37]. As expected, UUO led to significant fibrotic changes in the obstructed kidneys at 5 and 15 days after the operation (Fig. 4a, b and d–g). Remarkably, UUO induced a marked decrease of SNCA mRNA in the obstructed kidneys starting from day 5 post-UUO (Fig. 4c), subsequently leading to a further decrease with the progression of the disease (day 15 post-UUO, Fig. 4c). Immunohistochemistry study showed that SNCA protein was mainly expressed in the cytoplasm and nuclei of the healthy renal tubular epithelium (Fig. 4g and Supplementary Fig. 4, A, B). Collecting ducts stained positively, as did glomeruli (Supplementary Fig. 4 C, D). After challenge with UUO, mice exhibited an evident decrease of SNCA expression in dilated renal tubules as early as 5 days after surgery (Fig. 4g). Subsequently, levels of SNCA further decreased and almost disappeared in some tubules by day 15 after UUO (Fig. 4g). Furthermore, using triple immunofluorescence staining, we managed to identify SNCA negative tubular cells in areas of fibrosis showing de novo expression of vimentin, while still preserving the expression of epithelial marker E-cadherin (Supplementary Fig. 5A–D). Our results in vivo confirm a newly established concept of partial EMT program and its contribution to the development of renal fibrosis.

**SNCA expression decreases in human kidneys with fibrosis**. Next, we assessed the expression of SNCA in kidneys of patients with chronic kidney disease. As a control we used the healthy area of renal tissue from patients who underwent nephrectomy due to malignancies or hydronephrosis. Paraffin-embedded renal tissue sections were stained for SNCA, α-SMA, FSP1, and Sirius Red. Similar to mouse kidney, SNCA was predominantly localized in the cytoplasm and nuclei of the healthy renal tubular epithelium (granular staining) (Fig. 5a). As expected, chronic kidney disease (CKD) samples showed significant levels of interstitial fibrosis (Fig. 5a, c–d) (see patient's characteristics in Supplementary Table 1). We found that, in patient samples, SNCA expression significantly decreased in the cytoplasm and nuclei of the dilated renal tubules (Fig. 5a, b) Linear regression analysis showed an inverse correlation between the percentage of SNCA positive tubules and the degree of α-SMA staining, as well as the extent of renal interstitial fibrosis measured by Sirius red staining (Fig. 5e and f, respectively). We did not detect any correlation between the tubular SNCA expression and the glomerular filtration rate (eGFR) ($R^2 = 0.0204$; $p = 0.3862$). Additionally, we were able to find dilated tubules in the areas of fibrosis with tubular cells showing absence of SNCA expression while presenting de novo expression of mesenchymal marker vimentin (Supplementary Fig. 6).

**RPTEC-specific SNCA gene deletion in mice**. To investigate the physiological function of proximal tubule SNCA and its role in the pathogenesis of renal fibrosis, we generated mice with specific deletion of SNCA from renal proximal tubule by crossing homozygous SNCA-floxed mice (SNCA$^{flox}$) with PEPCK$^{Cre+}$ transgenic mice. PCR analysis of tail DNA confirmed the genotypes of proximal tubule specific SNCA null mice (PEPCK$^{Cre+}$-SNCA$^{fl/fl}$) and the control wild-type mice (PEPCK$^{Cre+}$-SNCA$^{wt/wt}$) (Supplementary Fig. 7 A, B). Bands of WT or floxed SNCA alleles were observed at ~390 bp and 440 bp, respectively (Supplementary Fig. 7A), while the expression of Cre transgene was visualized by a 465 bp PCR product (Supplementary Fig. 7B). PEPCK$^{Cre+}$-SNCA$^{fl/fl}$ (mutant) mice were born at the expected Mendelian ratio and did not show abnormalities at birth, neither at the age of 8–10 weeks when the experiments started.

To affirm specific Cre recombinase activity, we crossed PEPCK$^{Cre}$ mice with reporter mT/mG [B6.129(Cg)-Gt(ROSA) 26Sor$^{tm4(ACTB-tdTomato-EGFP)Luo/J}$] mice. Supplementary Figure 7C demonstrates kidney EGFP reporter immunofluorescence expression representative of Cre recombinase activity in B6.129(Cg)-Gt (ROSA)26Sor$^{tm4(ACTB-tdTomato-EGFP)Luo/J}$ mouse crossed with PEPCK$^{Cre+}$ mouse. The specificity of SNCA deletion in renal proximal tubular epithelial cells (RPTEC-SNCA deletion) was assessed by double immunofluorescence staining of SNCA with lotus tetragonolobus agglutinin (LTA), a specific renal proximal tubule marker[38,39], in kidneys of PEPCK$^{Cre+}$-SNCA$^{wt/wt}$ and PEPCK$^{Cre+}$-SNCA$^{fl/fl}$ mice at baseline (Fig. 6a, b). While the expression of LTA on the brush border of proximal tubules stayed unchanged in both groups of mice, SNCA expression on RPTECs of PEPCK$^{Cre+}$-SNCA$^{fl/fl}$ mice was visibly reduced, and nearly absent in certain areas (Fig. 6a). The SNCA protein expression pattern remained normal in the distal tubular cells (Fig. 6a). Likewise, the mRNA expression of SNCA in the whole kidney was significantly reduced (t-test: $p < 0.05$, Fig. 6c). These results affirmed SNCA deletion from RPTECs of the mutant mice.

**RPTEC-SNCA deletion accelerates the profibrotic gene expression after UUO**. Next, we subjected PEPCK$^{Cre+}$-SNCA$^{wt/wt}$ and PEPCK$^{Cre+}$-SNCA$^{fl/fl}$ mice to UUO challenge to determine if SNCA deficiency from RPTECs may render the obstructed kidneys of mutant mice more susceptible to renal fibrosis. mRNA expression of type I collagen, fibronectin, α-SMA and vimentin increased in obstructed kidneys of both PEPCK$^{Cre+}$-SNCA$^{wt/wt}$ and PEPCK$^{Cre+}$-SNCA$^{fl/fl}$ mice as early as 5 days after UUO, but to a significantly higher extent in PEPCK$^{Cre+}$-SNCA$^{fl/fl}$ mice (Fig. 6d; two-way ANOVA: $p < 0.001$, $p < 0.01$, $p < 0.05$, respectively). Of note, expression of α-SMA, vimentin, type I collagen and TGF-β1 mRNA showed a modest increase in the contralateral kidneys of PEPCK$^{Cre+}$-SNCA$^{fl/fl}$ mice 5 days after UUO, compared with control littermates (Fig. 6d), reaching statistical significance 15 days after UUO (Fig. 6e). As the disease progressed, obstructed kidneys of mutant mice still maintained the higher expression levels of profibrotic genes compared with control littermates, with collagen I mRNA reaching statistical significance (Fig. 6e). Of note, levels of cadherin-16 mRNA, a kidney-specific cadherin expressed in renal epithelial cells[9], were significantly lower in obstructed kidneys of PEPCK$^{Cre+}$-SNCA$^{fl/fl}$ animals 15 days after UUO, compared with PEPCK$^{Cre+}$-SNCA$^{wt/wt}$ mice (Fig. 6e).

**RPTEC-SNCA deletion increases interstitial matrix deposition and myofibroblast activation after UUO**. UUO led to a marked increase in collagen accumulation and deposition in wild-type kidneys at both time points (Fig. 7k, l, o, p), while disruption of SNCA gene from RPTECs further enhanced the collagen deposition seen as a greater red staining area in kidneys of PEPCK$^{Cre+}$-SNCA$^{fl/fl}$ mice at day 15 after UUO (Fig. 7k, l, o, p). Immunohistochemistry for detection of collagen type I in mouse

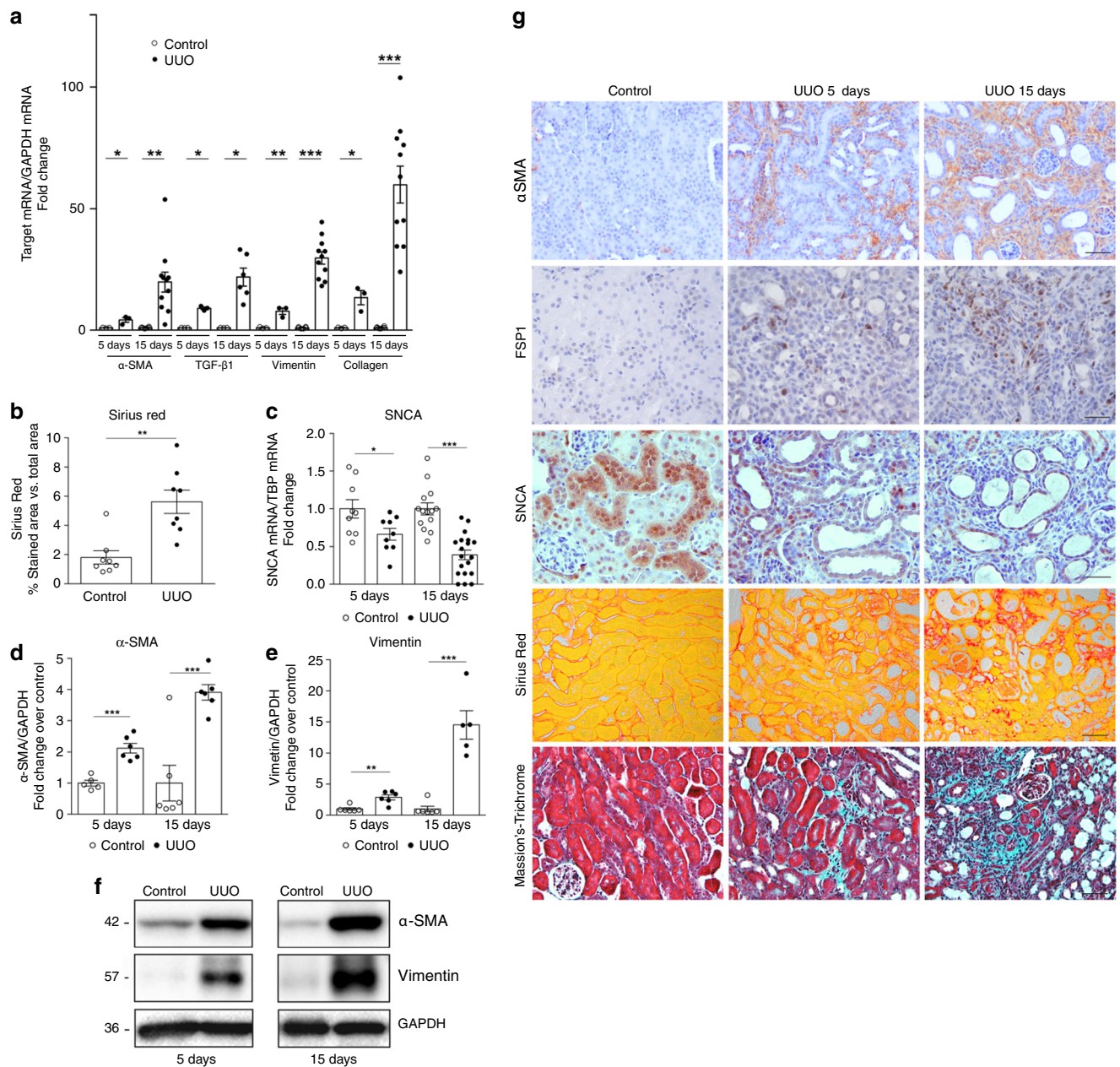

**Fig. 4 SNCA decreases and fibrotic markers increase in obstructed mouse kidneys. a**, **c** Total mRNA was extracted from kidneys and mRNA levels were determined by real-time qPCR. Relative mRNA levels were calculated and expressed as fold change over contralateral controls (value = 1.0) after normalizing for GAPDH or TBP. Data are presented as mean ± SEM (**a**: $n = 8$-11 (UUO15) or 3–5 (UUO5) mice/group; **c**: $n = 14$–19 (UUO15) or $n = 9$ (UUO5) mice/group). **b** Quantification of collagen content after Sirius red staining was expressed as a positive stained area vs. total analyzed area. Data are presented as mean ± SEM ($n = 8$ mice/group). **f** Representative Western blot analysis of α-SMA and vimentin in the kidneys of mice subjected to UUO. **d**, **e** Quantitative analysis by densitometry. Data were normalized to GAPDH and presented as mean ± SEM ($n = 8$ mice/group) (fold change over contralateral control). **g** Representative micrographs illustrating the expression of α-SMA, FSP1 and SNCA in mouse kidneys after UUO. Scale bar represents 50 μm. Masson´s trichrome and Sirius Red staining's depict collagen deposition (blue or red, respectively) in kidneys affected by fibrosis. *$p < 0.05$, **$p < 0.01$, ***$p < 0.001$ vs. contralateral control. The $p$-value by two-tailed Student's -test. Source data are provided as a Source Data file.

kidneys corroborated the results obtained after Sirius red staining (Fig. 7m–p). Consistently, the collagen I protein levels were significantly higher in PEPCK[Cre+]-SNCA[fl/fl]-obstructed kidneys than in wild-type obstructed kidneys at day 15 of UUO (Fig. 7b and j; two-way ANOVA: $p < 0.05$). Fibronectin protein levels showed a significant increase in PEPCK[Cre+]-SNCA[fl/fl]-obstructed kidneys at day 5 after UUO, compared with wild-type littermates (Fig. 7a, e). The protein expression of α-SMA and vimentin, molecular markers of mesenchymal phenotype, was notably

induced by UUO in kidneys of both PEPCK[Cre+]-SNCA[wt/wt] and PEPCK[Cre+]-SNCA[fl/fl] mice (Fig. 7a–d, g–h), however a significantly stronger expression of both myofibroblast markers was observed in the kidneys of mutant mice at day 5 (α-SMA, vimentin) and day 15 (α-SMA) of UUO (Fig. 7a–d, g–H). Consistently, immunohistochemistry showed more prominent staining for α-SMA and fibroblast specific protein (FSP1) in the obstructed kidneys of mutant mice compared with their control littermates (Fig. 7o, p).

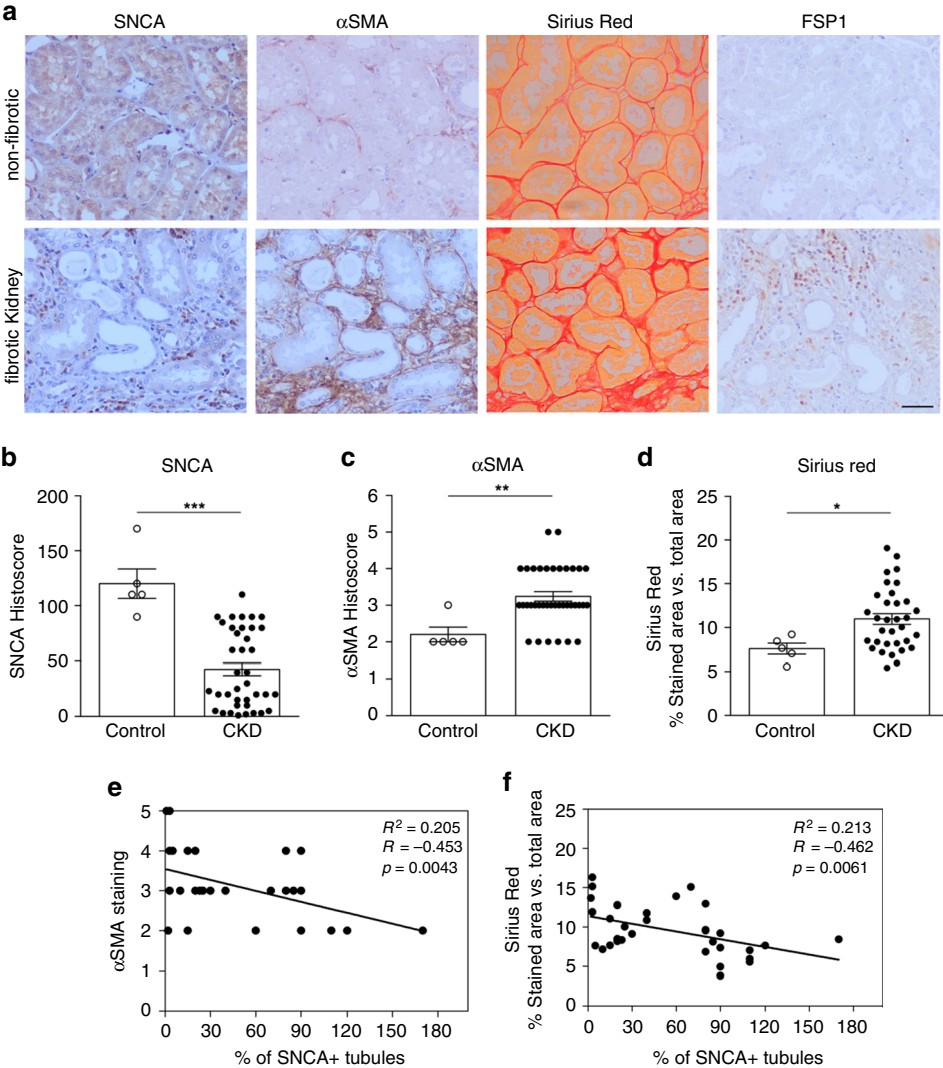

**Fig. 5 Decreased SNCA expression in human fibrotic kidney samples. a** Representative images of immunoperoxidase staining for SNCA, α-SMA, FSP1, and Sirius Red staining in human kidney sections. Scale bar represents 50 μm. **b**, **c** Quantification of SNCA staining in tubular cells (**b**) and α-SMA immunostaining (**c**) in kidney sections of patients with proven tubulointerstitial fibrosis versus kidney sections with no evidence of fibrosis. Data are presented as mean ± SEM ($n = 5$, healthy tissue; $n = 38$, fibrotic tissue). **d** Quantification of collagen content after Sirius red staining was shown as a positive stained area vs. total analyzed area. Data are presented as mean ± SEM ($n = 5$, healthy tissue; $n = 38$, fibrotic tissue). **e**, **f** Scatter plots with linear regression show correlation analysis between the percentage of SNCA positive tubules and **e** the degree of α-SMA staining and/or **f** the extent of renal fibrosis evaluated by Sirius red staining. Linear regression shows an inverse correlation between the percentage of SNCA positive tubules and the degree of α-SMA staining (**e**), as well as the fibrosis score (**f**). The Spearman correlation coefficient ($R$) and $p$ value are shown. *$p < 0.05$, **$p < 0.005$, ***$p < 0.001$ vs. control healthy kidney tissue. The p-value by two-tailed Student's $t$-test. Source data are provided as a Source Data file.

**RPTEC-SNCA deletion aggravates adenine-induced renal fibrosis.** To validate our findings, we extended our studies to an additional model of renal fibrosis, adenine-induced tubulointerstitial nephropathy in mice. Of note, four weeks of adenine diet feeding led to a development of tubular interstitial fibrosis and the significant decline of SNCA mRNA in the kidneys of WT mice, compared with wild-type littermates fed a normal diet (Fig. 8k). Furthermore, adenine diet induced a significant increase of blood urea nitrogen (BUN) in the serum of both PEPCK$^{Cre+}$-SNCA$^{wt/wt}$ and PEPCK$^{Cre+}$-SNCA$^{fl/fl}$ mice, compared with their counterparts fed a regular diet (Supplementary Fig. 8). As expected, adenine rich diet induced an increase of profibrotic markers such as type I collagen, fibronectin, α-SMA, vimentin and TGF-β1 mRNA and a decrease of cadherin-16 mRNA in kidneys of PEPCK$^{Cre+}$-SNCA$^{wt/wt}$ mice compared with wild-type littermates fed a normal diet (Fig. 8a–f). Notably, levels of tested profibrotic

genes were significantly higher in PEPCK$^{Cre+}$-SNCA$^{fl/fl}$ mice fed an adenine diet (Fig. 8a–f, two-way ANOVA: $p < 0.01$ Coll I, α-SMA, FN; $p < 0.05$, vimentin), except for TGF-β1, which did not reach statistical significance (Fig. 8e). Adenine rich diet led to a marked increase of collagen accumulation and deposition in wild-type kidneys (Fig. 8j, n), while disruption of SNCA gene from RPTECs further enhanced the collagen deposition seen as a greater red staining area in kidneys of PEPCK$^{Cre+}$-SNCA$^{fl/fl}$ mice fed an adenine rich diet (Fig. 8j, n). Immunohistochemistry for detection of collagen type I in mouse kidneys corroborated the results obtained after Sirius red staining (Fig. 8l, n, two-way ANOVA: $p < 0.01$). The protein expression of α-SMA, vimentin and collagen I was notably induced by adenine rich diet in kidneys of both PEPCK$^{Cre+}$-SNCA$^{wt/wt}$ and PEPCK$^{Cre+}$-SNCA$^{fl/fl}$ mice (Fig. 8g–i, m), however a significantly stronger expression of all tested markers was observed in the kidneys of mutant mice fed

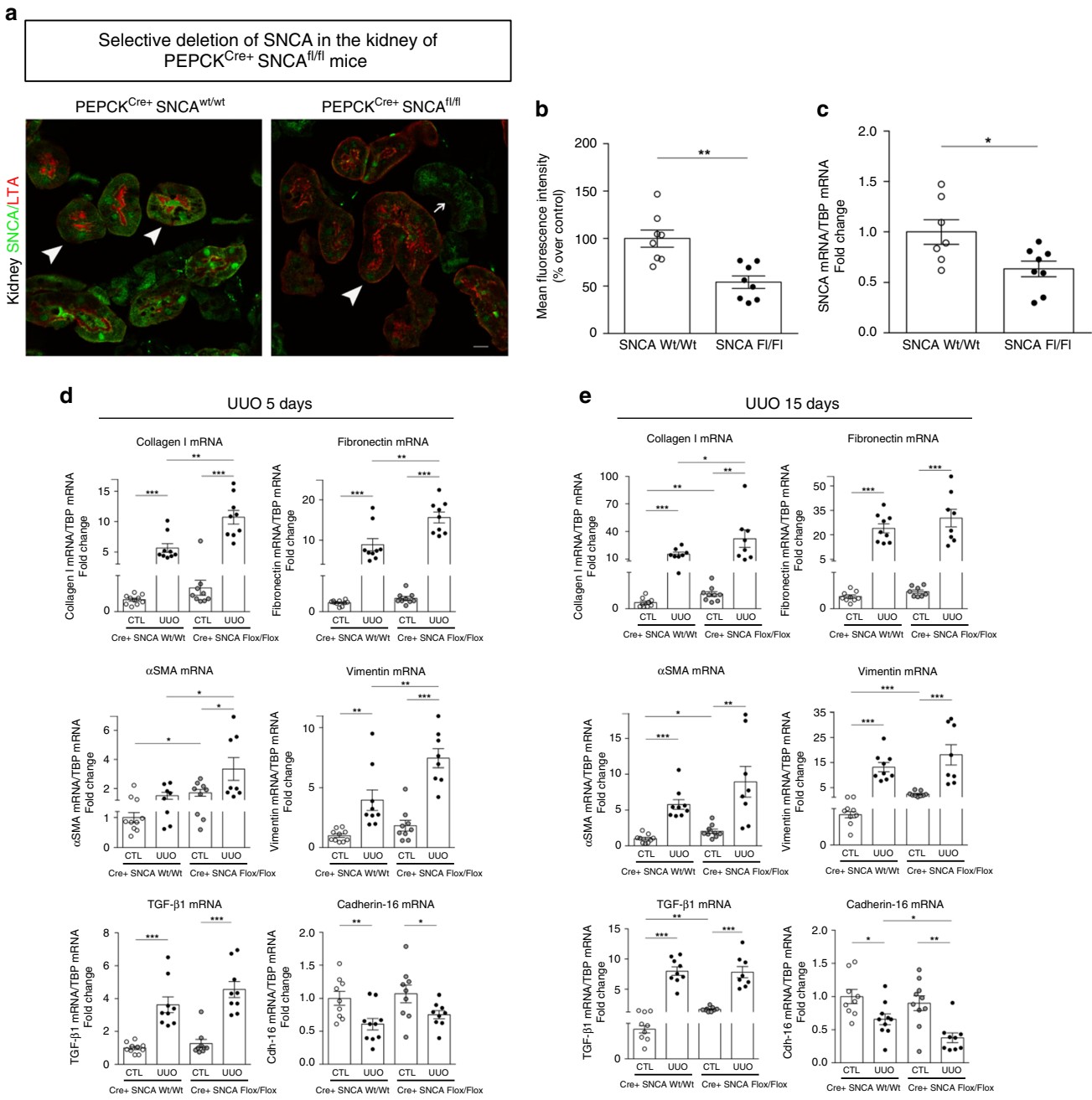

**Fig. 6 RPTEC-SNCA deletion accelerates the profibrotic gene expression after UUO. a** Specificity of SNCA knockdown in renal proximal tubular epithelial cells was confirmed by double immunofluorescence staining of SNCA (green) with lotus tetragonolobus agglutinin (LTA, red) in PEPCK[Cre+] SNCA[wt/wt] and PEPCK[Cre+] SNCA[fl/fl] mouse kidneys at baseline. LTA is a marker of RPTECs and no difference in its expression was observed between WT and Flox mice (thick arrows, LTA-positive tubules; thin arrow, LTA-negative tubules). SNCA expression on RPTECs of PEPCK[Cre+] SNCA[fl/fl] mice was dramatically reduced, and nearly absent in some areas. Representative images are shown for each genotype. Scale bar represents 10 µm. **b** Quantification of SNCA positively stained areas of proximal tubules from PEPCK[Cre+] SNCA[wt/wt] and PEPCK[Cre+] SNCA[fl/fl] mouse kidneys at baseline. Data are presented as mean ± SEM ($n = 8$ mice/group) (% over control). **c** Total mRNA was extracted from kidneys of PEPCK[Cre+] SNCA[wt/wt] and PEPCK[Cre+] SNCA[fl/fl] mice. mRNA levels of SNCA were determined by quantitative real-time PCR and normalized to TBP. Data are presented as mean ± SEM ($n = 8$ mice/group). **d–e** Total mRNA was extracted from contralateral non-obstructed (CTL) and obstructed (UUO) kidneys of PEPCK[Cre+] SNCA[wt/wt] and PEPCK[Cre+] SNCA[fl/fl] mice 5 days (**d**) and 15 days (**e**) after UUO. mRNA levels of collagen I, fibronectin, α-SMA, vimentin, TGF-β1, and cadherin-16 were determined by quantitative real-time PCR and normalized to TBP. Data are presented as mean ± SEM ($n = 10–15$ mice/group). *$p < 0.05$, **$p < 0.01$, ***$p < 0.001$. The $p$-value by two-tailed Student's $t$-test (**b**, **c**) or two-way ANOVA (**d**, **e**). RPTEC—renal proximal tubular epithelial cell. Source data are provided as a Source Data file.

an adenine diet (Fig. 8g–i, m; two-way ANOVA: $p < 0.01$ α-SMA; $p < 0.05$ Coll I). Consistently, immunohistochemistry showed more prominent staining for α-SMA and FSP1 in the kidneys of mutant mice fed an adenine rich diet compared with PEPCK[Cre+]-SNCA[wt/wt] littermates fed the same diet (Fig. 8n).

Furthermore, triple immunofluorescence staining of kidney sections from wild-type mice fed an adenine rich diet revealed that various tubular cells in the areas of fibrosis showing decreased or even lost SNCA expression begin to express mesenchymal marker vimentin, while still preserving the

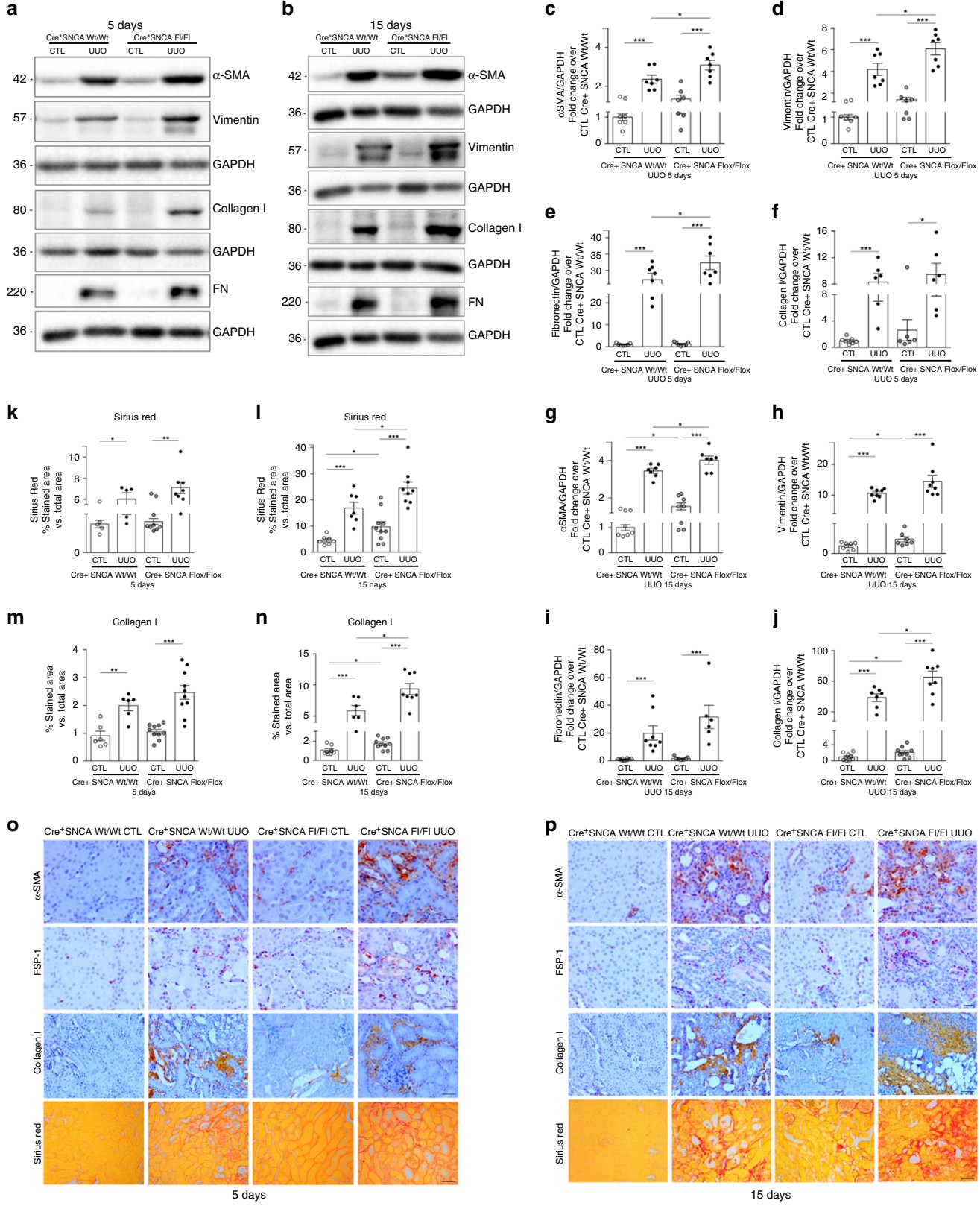

expression of epithelial marker E-cadherin (Supplementary Fig. 9a–d).

**SNCA modulates the activation of ERK1/2, Akt, and p38 after UUO.** To assess the involvement of MAPK and PI3K-Akt pathways in accelerated fibrotic response in PEPCK[Cre+]-SNCA[fl/fl]

mice upon UUO in vivo, we analyzed the expression levels of pERK1/2, pAkt, and p-p38 in the kidneys of both PEPCK[Cre+]-SNCA[wt/wt] and PEPCK[Cre+]-SNCA[fl/fl] mice. We found a marked increase of pERK1/2, pAkt, and p-p38 in obstructed kidneys of PEPCK[Cre+]-SNCA[fl/fl] animals as early as 5 days after UUO, compared with PEPCK[Cre+]-SNCA[wt/wt] mice, with p-p38

**Fig. 7 RPTEC-SNCA deletion increases interstitial matrix deposition and myofibroblast activation after UUO.** PEPCK[Cre+] SNCA[wt/wt] and PEPCK[Cre+] SNCA[fl/fl] mice were subjected to UUO and sacrificed 5 and 15 days after obstruction. **a, b** Whole kidney lysates from contralateral non-obstructed (CTL) and obstructed (UUO) kidneys were processed for protein analysis at days 5 and 15 after UUO and were immunoblotted with antibodies against α-SMA, vimentin, collagen I, fibronectin and GAPDH. **a, b** Representative Western blot analysis of α-SMA, vimentin, collagen, and fibronectin in the kidneys of PEPCK[Cre+] SNCA[wt/wt] and PEPCK[Cre+] SNCA[fl/fl] mice subjected to UUO. **c–j** Quantitative analysis by densitometry. Data were normalized to GAPDH and presented as mean ± SEM ($n = 7$ mice/group for UUO5; $n = 7$–9 mice/group for UUO15) (fold change over CTL PEPCK[Cre+] SNCA[wt/wt]). Quantification of collagen deposition in kidneys of PEPCK[Cre+] SNCA[wt/wt] and PEPCK[Cre+] SNCA[fl/fl] mice after Sirius red staining (**k, l**) and immunohistochemistry for collagen I (**m, n**) was expressed as a positive stained area vs. total analyzed area. Data are presented as mean ± SEM ($n = 5$–9 mice/group (UUO5), $n = 7$–10 mice/group (UUO15), Sirius red; $n = 6$–10 mice/group, collagen I). *$p < 0.05$, **$p < 0.01$, ***$p < 0.001$. The $p$-value by two-way ANOVA. **o, p** Representative images of Sirius red staining and immunohistochemistry for α-SMA, FSP1 and collagen I in mouse kidneys 5 (**o**) and 15 (**p**) days after UUO. Scale bar represents 50 μm (Collagen I, Sirius red) or 20 μm (α-SMA, FSP1). Source data are provided as a Source Data file.

and pAkt reaching statistical significance (Fig. 9a, c, d; two-way ANOVA: $p < 0.05$ pAkt). Furthermore, using double immunofluorescence staining, we managed to identify SNCA negative tubular cells in the areas of fibrosis showing increased expression of p-p38, while healthy SNCA positive tubules showed no staining for phosphorylated p38 (Supplementary Fig. 10a–c).

## Discussion

In this study, we describe the presence of SNCA in RPTECs and establish the functional significance of endogenous SNCA expression in these cells, proposing a role for SNCA in protecting kidney against fibrotic injury.

Renal fibrosis is a final manifestation of chronic kidney disease where TGF-β1 is considered a key mediator of fibrotic signaling in renal epithelial cells[34–36]. First, we sought to investigate the relationship between SNCA and TGF-β1 in vitro. Treatment with TGF-β1 led to marked changes in epithelial phenotype of HK-2 cells, which was in accordance with previously published results[32]. Furthermore, TGF-β1 led to a significant decrease of SNCA mRNA and protein expression in a dose- and time-dependent manner, suggesting a possible involvement of this protein in EMT of tubular epithelial cells. To corroborate the potential role of SNCA in the maintenance of the epithelial phenotype of renal tubular cells in vitro, the expression of SNCA in HK-2 was disrupted by shRNA delivery. Thus, knockdown of SNCA in HK-2 caused a modest decrease of E-cadherin and a marked increase of the expression of mesenchymal markers, α-SMA and vimentin. The already increased expression levels of mesenchymal markers in shSNCA cells did not further increase with TGF-β1 treatment, suggesting an overriding effect of SNCA, at least in vitro. On the other hand, overexpression of SNCA in HK-2 cells managed to blunt the TGF-β1-induced expression of major cytoskeletal components of mesenchymal cells[40], α-SMA and vimentin. The role of SNCA in regulating actin cytoskeletal organization has been previously reported in neurons[27–29], emphasizing its function as a major modulator of microfilament function and cytoskeletal dynamics[27–29]. Thus, Sousa et al.[27] demonstrated that SNCA binds actin, slows down its polymerization and accelerates its depolymerization. Furthermore, Bellani et al.[29] indicated that SNCA was able to reduce the pool of polymerized actin available in the cells by sequestering actin monomers. Results of our study show that knockdown of SNCA in HK-2 cells stimulated reorganization of the actin cytoskeleton seen as a stress fiber formation. In our study, we did not detect an increase of other synuclein's oligomers in HK-2 cells carrying shRNA for SNCA, but we still cannot absolutely exclude possible toxicity of these structures even in picomolar concentrations. Further investigation is needed to elucidate the possible formation and functionality of oligomeric SNCB and SNCG in the settings of SNCA deficiency in HK-2 cells.

Once established that the expression of SNCA was decreased in TGF-β1-treated cultures, we assessed the expression of SNCA in a

mouse model of renal fibrosis, as well as in patients with kidney disease. Our results demonstrate that SNCA was significantly downregulated in dilated renal tubules of both human and mouse kidneys affected by fibrosis, compared with normal human kidney tissue or contralateral non-obstructed mouse kidneys in the UUO model, respectively. These findings suggest to a dysregulation of endogenous SNCA levels in kidney fibrosis and raise the possibility that SNCA could play a role in fibrosis development. In our human study, we detected an inverse correlation between the percentage of SNCA positive tubules and the degree of α-SMA and Sirius red staining in fibrotic kidneys; however, there was no correlation between the tubular SNCA expression and the eGFR. The lack of association between the loss of tubular SNCA expression and declining renal function presumably reflects the rather eclectic group of CKD patients analyzed in our study, with only small number of patients with glomerulonephritis.

To circumvent the systemic effect of global SNCA depletion using a conventional SNCA knockout mouse model, and to directly assess the potential role of renal proximal tubule SNCA in kidney fibrosis, we used conditional gene targeting and selectively deleted SNCA expression in mouse RPTECs. We found that PEPCK[Cre+]-SNCA[fl/fl] mice developed more severe tubulointerstitial injury and fibrosis after UUO, compared with control littermates. Thus, normal endogenous expression of SNCA in RPTECs significantly attenuated the detrimental effects of obstructive nephropathy. The protective effect of SNCA seems to involve a reduction of profibrotic gene expression, as well as decreased myofibroblast activation and interstitial matrix deposition. Myofibroblasts are considered the principal mediators of renal fibrosis responsible for the interstitial matrix accumulation and deposition[41,42]. Indeed, we observed significantly higher mRNA and protein expression of α-SMA and vimentin in the obstructed kidneys of PEPCK[Cre+]-SNCA[fl/fl] mice. Furthermore, SNCA expression also reduced ECM production in obstructed kidneys. Namely, PEPCK[Cre+]-SNCA[fl/fl] mice showed significantly higher levels of fibronectin mRNA expression as well as collagen I mRNA and protein levels, which led to an aggravation of the fibrotic phenotype after UUO. Importantly, our in vivo results were confirmed in a second model of renal fibrosis, adenine-induced tubulointerstitial nephropathy in mice. Results obtained using two mechanistically distinct mouse models of renal fibrosis support the finding that SNCA from RPTECs has a role in protecting kidney parenchyma against injury. Several studies demonstrated a protective effect of SNCA against different types of injuries in the brain[20,23,25,26,43], including the CSPα depletion-induced neurodegeneration that led to gliosis[23], a form of fibrosis of the CNS[44].

The MAPK & PI3K pathways have been reported to play a vital role in the EMT of different epithelial cells in vitro[32,45–47], as well as the progression of renal fibrosis in vivo[36]. In the present study, SNCA knockdown led to a significant increase of pAkt and p-p38 in HK-2 cells. The already elevated expression levels of pAkt and

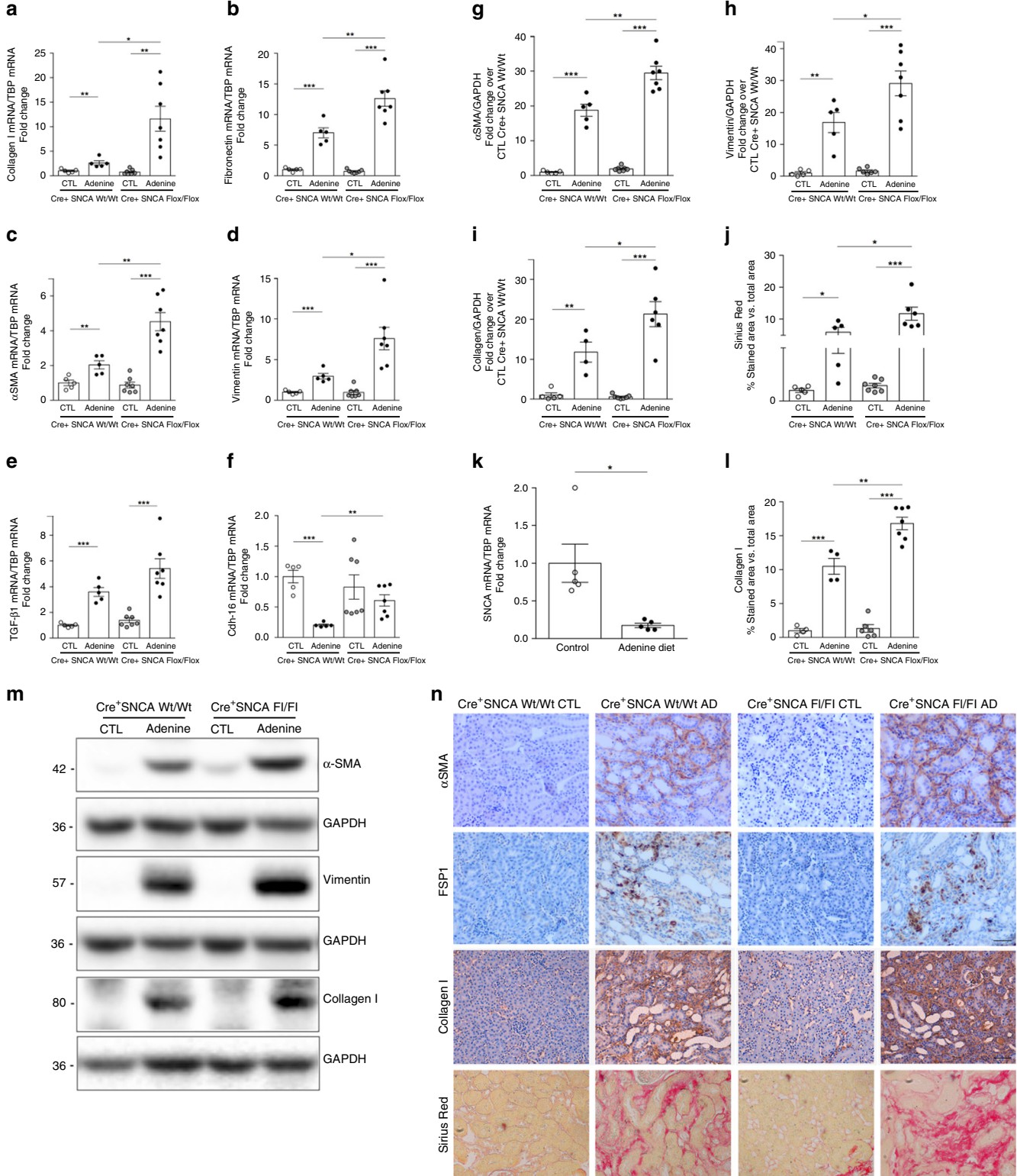

**Fig. 8 RPTEC-SNCA deletion aggravates adenine-induced renal fibrosis. a–f, k** Total mRNA was extracted from kidneys of PEPCK[Cre+] SNCA[wt/wt] and PEPCK[Cre+] SNCA[fl/fl] mice fed a regular or adenine rich diet for 4 weeks. mRNA levels of collagen I, fibronectin, α-SMA, vimentin, TGF-β1, cadherin-16 and SNCA were determined by quantitative real-time PCR and normalized to TBP. **m** Whole kidney lysates were processed for protein analysis and were immunoblotted with antibodies against α-SMA, vimentin, collagen I and GAPDH. **m** Representative Western blot analysis and **g–i** quantitative analysis by densitometry. **j, l** Quantification of collagen deposition in kidneys of PEPCK[Cre+] SNCA[wt/wt] and PEPCK[Cre+] SNCA[fl/fl] mice after Sirius red staining (**j**) and immunohistochemistry for collagen I (**l**) was expressed as a positive stained area vs. total analyzed area. Data are presented as mean ± SEM ($n = 5$–7 mice/group). *$p < 0.05$, **$p < 0.01$, ***$p < 0.001$. The $p$-value by two-way ANOVA. **n** Representative images of Sirius red staining and immunohistochemistry for α-SMA, FSP1 and collagen I in mouse kidneys. Scale bar represents 50 μm. Source data are provided as a Source Data file.

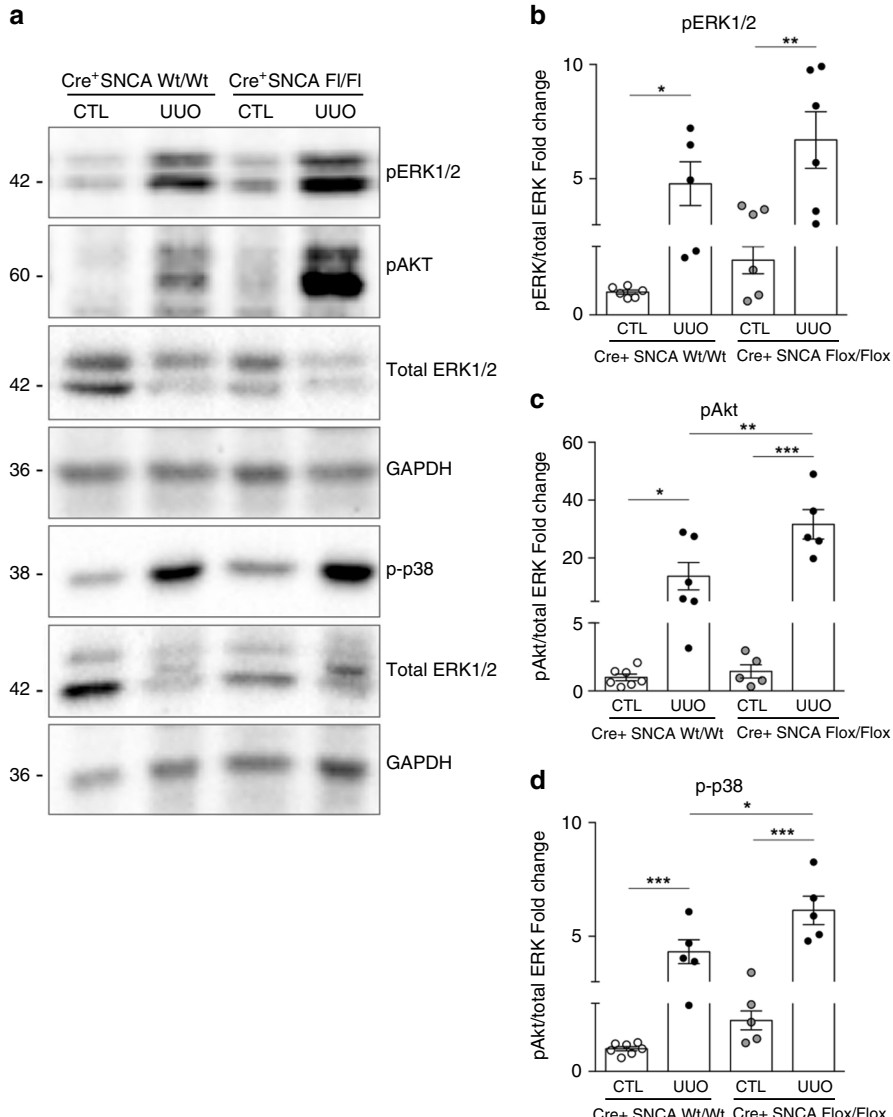

**Fig. 9 SNCA modulates the activation of ERK1/2, Akt and p38 after UUO.** PEPCK[Cre+] SNCA[wt/wt] and PEPCK[Cre+] SNCA[fl/fl] mice were subjected to UUO and sacrificed 5 days after obstruction. Whole kidney lysates from contralateral non-obstructed (CTL) and obstructed (UUO) kidneys were processed for protein analysis and were immunoblotted with antibodies against pERK1/2, pAkt, p-p38, total ERK1/2 and GAPDH. **a** Representative Western blots and quantitative densitometric analysis **b–d** levels of phosphorylated ERK1/2 (**a**, **b**), Akt (**a**, **c**) and p38 (**a**, **d**) in the kidneys of PEPCK[Cre+] SNCA[wt/wt] and PEPCK[Cre+] SNCA[fl/fl] mice subjected to UUO. Data were normalized to total ERK1/2 and presented as mean ± SEM ($n = 7$ mice/group) (fold change over CTL PEPCK[Cre+] SNCA[wt/wt]). *$p < 0.05$, **$p < 0.01$, ***$p < 0.001$. The $p$-value by two-way ANOVA. Source data are provided as a Source Data file.

p-p38 in shSNCA cells did not further increase with TGF-β1 treatment, implying to an outweighing effect of SNCA knockdown in vitro. On the other hand, SNCA overexpression did not show marked effects on the activity of Akt and p38 in basal state, nonetheless there was a tendency toward decreasing the levels of the two phosphorylated proteins after TGF-β1 treatment. Furthermore, our results in vitro were confirmed in vivo. Namely, PEPCK[Cre+]-SNCA[fl/fl] mice showed significantly higher levels of p-p38 and pAkt in obstructed kidneys, compared with PEPCK[Cre+]-SNCA[wt/wt] littermates. Wishing to further study the mechanism governing this activation of p38 and Akt by SNCA knockdown we found that SNCA associates and binds to p38 MAPK, while overexpression of SNCA decreases phosphorylation of p38 in cells with constitutive activation of MKK6 kinase. Based on the fact that SNCA binds to p38 MAPK, it is more likely that the loss of SNCA affects phosphorylation of this signaling molecule. The binding of SNCA to p38 could explain not only the activation of p38 upon SNCA downregulation, but also the subsequent activation of Akt in

HK-2 cells. Namely, it has been shown that p38 pathway can influence Akt signaling, placing Akt kinase downstream of p38 MAPK[48,49]. Cabane et al.[48] demonstrated that stimulation of p38 resulted in a simultaneous activation of Akt pathway, while the inhibition of p38 led to a reduction of pAkt levels. Interestingly, neither inhibition nor activation of Akt had any effect on p-p38 levels in C2C12 cells[48]. McGuire et al.[49] confirmed the results from previous group demonstrating the existence of a cross talk between the p38α-MK2/3 and Akt pathways in macrophages and described a role for p38α-MK2/3 in regulating TLR-induced Akt activation in macrophages. The results obtained here depict the role for SNCA in regulating MAPK-p38 and PI3K-Akt pathways and are in line with previously published data in CNS[21,22,50]. Namely, Iwata et al.[50] demonstrated that SNCA affects the phosphorylation of MAPKs and that binding of SNCA to p38 MAPK regulates MAPK pathway by reducing the amount of available p38 to be phosphorylated. Furthermore, Musgrove et al.[21] showed that SNCA, through inhibition of the MAPK signaling pathway, prevented cytochrome c

release and apoptosis. Our focus on MAPK-p38 and PI3K-Akt pathways in this study does not exclude the possibility of other signaling cascades mediating the effects of loss of SNCA in RPTECs. Namely, SNCA has an important role in mitochondrial homeostasis and the alteration of this protein could lead to bioenergetics defects and neuronal impairments[51]. Considering the fact that RPTECs represent an important energy-demanding cell type that contains more mitochondria than any other structure in the kidney, the potential role of SNCA in the mitochondrial bioenergetics of RPTECs could not be neglected. Further investigation in this direction is needed as it may provide more insights into the complex roles of SNCA in peripheral tissues such as the kidney.

To investigate the molecular mechanism underlying TGF-β1-induced decrease in SNCA expression in vitro, we performed MAPK and PI3K inhibitor study. We found that SB203580 significantly attenuated TGF-β1-mediated decrease in SNCA expression, while LY294002 managed to partially blunt it. It thus seems like that the decrease in SNCA expression induced by TGF-β1 is mediated by MAPK-p38. Similar MAPK-p38 and PI3K-dependent regulation of SNCA expression was reported by Gomez-Santos et al.[52] and Clough et al.[53]. In their reports, the increase in SNCA expression caused by dopamine[52] or NGF and bFGF[53] was attenuated by SB203580[52] or LY294002[53], respectively. The present study reports that TGF-β1 decreases SNCA expression via MAPK-p38 activation.

In conclusion, our study provides evidence for an important role of SNCA in the maintenance of the epithelial phenotype of RPTECs and in protecting kidney parenchyma against injury. Besides identifying the functional relevance for endogenous SNCA expression in the RPTECs and the kidney, our work also emphasizes the importance of preserving the basal SNCA levels in the kidney as a therapeutic strategy to attenuate the progression of kidney fibrosis.

## Methods

**Cell culture and treatments**. HK-2 cells (human renal proximal tubular epithelial cells)[54] (ATCC® CRL-2190™) were maintained in DMEM/F12 media (Gibco™) supplemented with 2% FBS, Hepes buffer, insulin, transferrin, sodium selenite, glucose, dexamethasone, EGF, penicillin, and streptomycin (Sigma Aldrich). Fresh growth medium was changed every 2–3 days. Before treatments, cells were growth arrested in serum-free medium and incubated separately with serum-free medium (control), transforming growth factor-β1 (TGF-β1: 1, 2, 4 ng/ml; R&D Systems, Minneapolis, MN) and/or U0126 (10 μM/L; #662005, Calbiochem), SB203580 (20 μM/L; #559389, Calbiochem), LY294002 (10 μM/L; #440202, Calbiochem) for different periods of time. Cells were maintained according to the described protocol, unless otherwise indicated.

**Lentiviral production and infection of HK-2 cells**. The shRNA vector was constructed by annealing complementary 60-mer oligonucleotides containing the 21-nucleotide target sequence in both the sense and antisense orientation separated by a 9-nt spacer. The 21-mer sequence to α-SYN (SNCA) was TGA-CAATGAGGCTTATGAAAT and is predicted to be specific only for α-SYN as determined by BLAST database searches. Oligonucleotides to produce shRNA were annealed in buffer (150 mM NaCl; 50 mM Tris, pH 7.6) and cloned into the AgeI-BamHI site of lentiviral vector for RNA interference-mediated gene silencing under the control of U6 promoter for the expression of short hairpin shRNAs and the Venus variant of GFP under the control of SV40 promoter for monitoring transduction efficiency. The pSIN-pgk-human Synuclein WT-WPRE plasmid (SIN-PGK-hsynuclein-WHV) for the production of lentiviral particles to express human SNCA was a kind gift from Dr. Bernard Schneider (Swiss Federal Institute of Technology Lausanne, Switzerland). To produce lentiviral particles, 293T cells were co-transfected by the polyethylenimine method with virion packaging elements (VSV-G and Δ8.9) and the shRNA producing vector. 293 T cells were allowed to produce lentiviral particles for 3 days in DMEM supplemented with 10% FBS, sodium pyruvate, nonessential amino acids, penicillin, and streptomycin. Culture medium was collected and centrifuged at 3000 rpm for 10 minutes. Supernatant was collected and filtered using Sartorius filters at 4000 rpm for 1 h, 4 °C. Filtered supernatant was added to the growing culture of HK-2 cells and incubated overnight. Next morning, fresh medium was replaced, and the cells were grown for an additional 3–4 days to allow endogenous gene knockdown. Western blot and real-time PCR were performed to check for α-SYN gene knockdown and/or overexpression.

**RNA purification and quantitative real-time PCR**. Total RNA was extracted from cultured cells or from whole kidney tissue using TRIzol reagent (Sigma, Madrid, Spain), and reverse transcription was performed with First Strand cDNA Synthesis Kit (AMV) (Roche) according to manufacturer's instructions. Real-time PCR with gene-specific TaqMan probes was performed with a CFX Real-Time PCR detection system (Bio-Rad Laboratories, Madrid, Spain) using TaqMan Universal PCR Master Mix, No AmpErase UNG. Forty cycles at 95 °C for 15 s and 60 °C for 1 min were performed[32,55–57]. Relative mRNA levels were calculated by standard formulae (ΔΔCt method) using GAPDH or TBP as an endogenous control. The results referred to a randomly selected basal sample considered as value = 1.0. Gene-specific TaqMan probes used in this study are indicated in the Supplementary Methods.

**Patients and human kidney samples**. Human kidney tissue samples were obtained from 19 patients who underwent nephrectomy due to malignancies or hydronephrosis, and 24 patients who were submitted to renal biopsy and were confirmed with IgA nephropathy or glomerulonephritis in the University Hospital Arnau de Vilanova in Lleida, between 2013 and 2018. The human kidney samples were obtained with the support of IRBLleida Biobank (B.0000682) and PLATA-FORMA BIOBANCOS PT17/0015/0027. Two experienced pathologists evaluated the samples for the presence of renal fibrosis.

The study involving human samples was approved by the Ethics Committee for the Clinical Investigation of the University Hospital Arnau de Vilanova in Lleida (CEIC-1587), and complied with all relevant ethical regulations and the guidelines of the Declaration of Helsinki. Prior to inclusion in the study, all patients provided an informed consent for the collection and use of their kidney tissues for research. The informed consent specified that medical data such as age, sex and BMI could be shared and used for research.

**Animals and experimental protocol**. Male C57BL/6J mice (8–12-weeks old), weighing ~20–23 g, were purchased from Charles River (Barcelona, Spain). Mice were housed and maintained in a barrier facility, and pathogen-free procedures were used in all mouse rooms. Animals were kept in a 12-h-light/dark cycle at 22 °C with ad libitum access to food and water.

The male C57BL/6J mouse model of renal fibrosis was established by UUO. Under general anesthesia (Isoflurane), male mice (8–10-weeks old) were subjected to UUO by double-ligating the left ureter using 4-0 silk after a lateral abdominal incision. After surgery mice received pain medication (buprenorphine, 0.05 mg/kg, sc). Mice were euthanized at days 5 and 15 after the surgery. Blood was collected by cardiac puncture and the animals were perfused with PBS through a puncture in the left ventricle. The organs of interest were collected for histologic examination and molecular analysis. One part of the kidney was fixed in 4% paraformaldehyde/PBS for histologic examinations after embedding in paraffin and/or Bright Cryo-M-Bed compound (Bright Instrument Co). The remaining kidney tissue was snap-frozen in liquid nitrogen and kept at −80 °C for protein and mRNA extractions.

All animal studies were approved by the local Animal Ethics Committee of the University of Lleida (CEEA 07-02/14), and complied with all relevant ethical regulations and the guidelines of European Research Council for the Care and Use of Laboratory Animals.

**Generation of RPTEC-specific SNCA null mice**. To generate RPTEC-specific SNCA null mice, SNCA^flox mice [B6(Cg)-Snca^tm1.1Vlb/J; JAX stock #025636] purchased from the Jackson Laboratory (Bar Harbor, ME)[58] were bred with a PEPCK^Cre+ transgenic mice (Cre recombinase under the control of the phosphoenolpyruvate carboxykinase promoter[39]) on a C57BL/6J background (kindly provided by Dr. Volker Haase, Vanderbilt University)[39] to yield PEPCK^Cre+-SNCA^wt/flox progeny. The PEPCK^Cre transgenic mice show ~10-fold increase in PEPCK expression in renal proximal tubule[59]. PEPCK^Cre+-SNCA^wt/flox mice were intercrossed to obtain the breeders to produce the experimental genotypes PEPCK^Cre+-SNCA^wt/wt (control group) and PEPCK^Cre+-SNCA^flox/flox (proximal tubular cell-specific SNCA deletion). Mice were genotyped by tail biopsy PCR using Cre transgene and SNCA^loxP specific primers (Supplemental Material). To assess the specificity of Cre-recombinase expression, PEPCK^Cre mice were crossed with reporter mT/mG [B6.129(Cg)-Gt(ROSA)26Sor^tm4(ACTB-tdTomato-EGFP)Luo/J] mice obtained from the Jackson Laboratory (Bar Harbor, ME). The mT/mG ROSA reporter mice express red fluorescence prior to, and green fluorescence following, Cre-mediated recombination. Male PEPCK^Cre+-SNCA^wt/wt and PEPCK^Cre+-SNCA^flox/flox mice (8–10 weeks old) were used in experiments of renal fibrosis induced by UUO.

**Statistical analyses**. Statistical analysis was performed using GraphPad Prism (GrahPad Software, San Diego California USA). All data are expressed as mean ± SEM. Differences among groups were assessed by one-way ANOVA or two-way ANOVA, followed up by Tukey's test, as needed. Differences between two groups were evaluated by the Student's t-test.

**Reporting summary**. Further information on research design is available in the Nature Research Reporting Summary linked to this article.

## Data availability

Source data are provided as a Source Data file. All other data supporting the findings of this study are included in the supplementary information or available from the corresponding author upon reasonable request.

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

## Acknowledgements

This work was supported by research grants PI15/00960 and PI17/00119 from the Instituto de Salud Carlos III (ISCII, Spanish Ministry of Economy and Competitiveness) and REDinREN (RD12/0021). The work on human kidney tissue samples was supported by the IRBLleida Biobank (B.0000682) and PLATAFORMA BIOBANCOS PT17/0015/0027. M.B. was supported by the REDinREN (RD12/0021/0026) and Department of Health, Government of Catalonia (PERIS 2016-2020, SLT002/16/00178). M.C. was supported by the studentship from the Catalan Government (AGAUR). R.R.R-D. was supported by a grant from the Comunidad Autonóma de Madrid (B2017/BMD-3751 NOVELREN-CM). We thank Dr. Bernard Schneider (Swiss Federal Institute of Technology Lausanne, Switzerland) for providing pSIN-pgk-human Synuclein WT-WPRE plasmid, Dr. Guadalupe Sabio (CNIC, Madrid, Spain) for providing HA-p38 PCNA plasmid, and Dr. Volker Haase (Vanderbilt University, TN, USA) for providing PEPCK-Cre+ transgenic mice. We thank Laura Santos-Sanchez (Universidad Autonoma Madrid) for helping with collagen evaluation and immunohistochemistry studies, and Alicia Garcia (IRBLleida) for valuable technical help in the laboratory.

## Author contributions

M.B. and J.M.V. conceived and designed the study; M.B., M.C., R.R.R-D., N.P., E.G., and A.M. carried out the experiments; M.B. analyzed the data and designed the figures; P.G. and M.J.P. performed an independent pathological evaluation of human kidney samples; M.B. wrote the manuscript and review, editing; M.R.O. and J.M.V. performed the critical review and editing of the manuscript; J.M.V., M.R.O., and E.F. obtained funding. All authors approved the final version of the manuscript.

## Competing interests

The authors declare no competing interests.
