## [Peer Review File · Nature Communications]

Reviewers' comments:

Reviewer #1 (Remarks to the Author):

The paper by Bozic et al illustrates a novel potential protective role of α -synuclein (SNCA) in the pathogenesis of renal fibrosis. The paper is well structured and the protective role of SNCA has been well evidenced. Since it is the first time that SNCA is proposed to have a role in the renal function some aspect should be clarified from the beginning. In general the information on the peripheral expression of SNCA in adult mammalian tissues is limited, the authors cite just a single paper (12)

1) As mentioned by the authors SNCA is part of a large family of proteins including β and γ synuclein, the presence of these two other proteins should be investigated in HK-2 cells, just to contribute to the specific effect of SNCA.

2) The involvement of SNCA in synucleinopathies is associated with its self-assembling capacity in small (oligomers) or large aggregates of the protein, can we exclude the formation of SNCA oligomers in the models or pathologies studied ?

3) A EM analysis of SNCA subcellular distribution will help to support the functional role of SNCA in the new context

4) Again in analogy with the role of SNCA in central nervous system the cell to cell passage of the protein could be investigated by the external application of SNCA to HK-2 cells

The term "tubular" for the general audience could be misinterpreted

Reviewer #2 (Remarks to the Author):

This is a potentially interesting study to investigate the role of alpha-synuclein (SNCA) in renal interstitial fibrosis. However, a number of significant concerns need to be addressed.

Major concerns:

1. The expression pattern of SCNA in mouse and human kidney should be described in greater detail. Of note, the Human Protein Atlas shows discrete staining of podocytes and no staining of tubular cells with two different SCNA antibodies. The data used to argue that a loss of SCNA expression is associated with renal fibrosis in human kidney disease is totally inadequate. The patients need to be characterised, SCNA and for SCNA and collagen/a-SMA staining quantified, and double staining or serial section staining much better illustrated.
2. While the in vitro studies argue that loss of SCNA expression leads to tubular cell de-differentiation towards a mesenchymal phenotype, this is not analysed in vivo. Do the SCNA negative tubular cells in areas of fibrosis show de novo expression of vimentin or Snail1 and a loss of E-cadherin and cytokeratin? This has not been investigated in either the human or mouse studies.
3. The efficiency of SCNA deletion in PEPCK expressing tubules has not been quantified.
4. The conclusion that loss of RPTEC expression of SCNA leads to greater renal fibrosis in the UUO model is not certain. Fig 6G shows no difference in Sirius red staining area in the day 5 obstructed kidney between the two genotypes. Fig 6H shows a 2-fold increase in collagen deposition in the contralateral control kidney on day 15 of UUO in the KO group – which was not evident at day 5 UUO. What has happened here? Do these mice spontaneously develop renal fibrosis with age? Also, this means that the fold difference in Sirius red staining between the obstructed and non-obstructed kidney in the KO mice (approx. 2.5 fold) is less than that seen in the WT control mice (approx. 4-fold). It would be helpful to show the actual area stained rather than fold change. In addition, analysis of deposition of a specific type of collagen would be helpful to validate the Sirius red staining. Most importantly, analysis of a second model of renal interstitial fibrosis is required to validate this possible finding.
5. Fig 5 – the Western blots for collagen I shown in Fig 5A & B do not concord with the graphs of the blots shown in Fig 5C & E. The blots indicate a substantial increase in collagen I content in the control kidney of mice with SCNA conditional deletion compared to the control kidney of non-deleted mice – but there is no difference in the blots. This does not correlate with the Sirius red staining data. Also, a 70kDa band is rather small for collagen I.
6. In Fig 3, why should knock-down of SCNA affect TGF- β 1 induced p38 or Akt signalling? How does this operate? Does over-expression of SNCA also affect these pathways?

Reviewer #3 (Remarks to the Author):

This manuscript described an interesting role for Alpha-(α)-synuclein (SNCA) in protecting against profibrotic changes of proximal tubular epithelial cells via inhibition of MAPK-p38 and PI3K-Akt pathways in EMT and ECM accumulation. It is well designed study with both in vitro and in vivo evidence to support their conclusions.

1. Morphological changes of HK-2 cells as evidence of TGF= β 1-induced EMT is required.
2. It is not clear what time points were samples analysed for E-cadherin, Vimentin, α -SMA in figure 1B. In time sequence of EMT in the in vitro study, is downregulation of the SNCA an early or late event.
3. Fig 2A does show a TGF= β 1-induced increase of α -SMA in shSNCA infested HK2 cells .
4. Fig 2B overexpression of SNCA showed a clear non-responsiveness of infested HK2 cells in TGF= β 1-induced increase of α -SMA and Vimentin. Would this result suggest a different mechanism to that of SNCA deficiency by shSNCA?
5. The study demonstrated for the first time that TGF= β down regulate SNCA via MAPK-p38 activation. What is the mechanism for the demonstrated SNCA inhibition of Akt and p38 activation which are both downstream of TGF= β as MPAK-p38.
6. in vitro study using HK2 cells may not support in vivo activation of myofibroblasts, as contribution of tubular epithelial cells towards myofibroblasts has been questioned by other researchers.

Reviewer #1 (Remarks to the Author):

The paper by Bozic et al illustrates a novel potential protective role of α -synuclein (SNCA) in the pathogenesis of renal fibrosis. The paper is well structured and the protective role of SNCA has been well evidenced. Since it is the first time that SNCA is proposed to have a role in the renal function some aspect should be clarified from the beginning. In general the information on the peripheral expression of SNCA in adult mammalian tissues is limited, the authors cite just a single paper (12).

Response: We thank the Reviewer for the positive assessment of our work.

Regarding the peripheral expression of SNCA in adult mammalian tissues, in the initial version of the manuscript, in the Introduction sections (page 3) after the phrase: "In addition to its distribution within the CNS, it has been reported that SNCA is also expressed in a variety of non-neuronal cells and tissues (12-17)", we cited references from 12 - 17. We have now revised the cited papers and updated the list of references for the revised version of the manuscript, as follows: Ltic et al. 2004 (PMID 14997013); Tong et al. 2010 (PMID 20457918), Lee et al. 2011 (PMID 21886844); Hashimoto et al. 1997 (PMID 9299413); Shin et al. 2000 (PMID 10774749); Shameli et al. 2016 (PMID 26517968); Nakai et al. 2007 (PMID 17475220).

1) As mentioned by the authors SNCA is part of a large family of proteins including β and γ synuclein, the presence of these two other proteins should be investigated in HK-2 cells, just to contribute to the specific effect of SNCA.

Response: We thank the Reviewer for rising this point that we also think it's important. We performed the qPCR analysis and immunofluorescence experiments showing expression levels of α -, β - and γ -synuclein in HK-2 cells in basal conditions. We detected significantly lower levels of β - and γ -synuclein mRNA in HK-2 cells compared with the levels of α -synuclein mRNA, which was confirmed by immunofluorescence analysis (**Supplementary Fig. 1A, B**). The results are now commented in the Result section.

2) The involvement of SNCA in synucleinopathies is associated with its self-assembling capacity in small (oligomers) or large aggregates of the protein, can we exclude the formation of SNCA oligomers in the models or pathologies studied?

Response: We acknowledge the reviewer point and we agree that synucleinopathies are related to the self-assembling of either mutated or highly expressed SNCA. However, in this case, the pathological state is associated with a decrease in the expression of endogenous SNCA, so the involvement of aggregates in the pathophysiology of the EMT is highly unlikely.

3) A EM analysis of SNCA subcellular distribution will help to support the functional role of SNCA in the new context.

Response: We thank the Reviewer for this valuable suggestion. We have now performed the transmission electron microscopy (TEM) analysis of mouse kidney sections for the subcellular distribution of SNCA. However, we have experienced a lot of problems with the antibodies, as they have not been used in EM before. We believe there was a problem

with the fixation, as for EM is much more denaturalizing than for regular IHC. In any case we have obtained some staining (in the cytoplasm of renal proximal tubular cells as in IHC), but with lower intensity. Results are attached for the reviewer's evaluation (**R Fig. 1**, attached at the end of this file).

4) Again in analogy with the role of SNCA in central nervous system the cell to cell passage of the protein could be investigated by the external application of SNCA to HK-2 cells.

Response: We thank the reviewer for this interesting suggestion. We have expanded the results of the overexpression of SNCA (instead of external application) and we have confirmed the effect inhibiting EMT. The use of external addition of SNCA or peptide mimetics with potentially lower toxicity is now being considered in further experiments to elucidate whether SNCA could be a target in delaying or even stopping renal fibrosis.

The term "tubular" for the general audience could misinterpreted.

Response: We agree with the reviewer and we have now clarified the term "tubular" in the revised version of the manuscript. Wherever the term "tubular" was found alone in the text, we enriched it with more information, such as renal proximal tubular epithelial cells (RPTECs), giving more details to the general audience of the type of cells we are talking about. In this context, even the title of the manuscript was changed from: "Protective role of tubular alpha-synuclein in" into "Protective role of renal proximal tubular alpha-synuclein in ...".

Reviewer #2 (Remarks to the Author):

This is a potentially interesting study to investigate the role of alpha-synuclein (SCNA) in renal interstitial fibrosis. However, a number of significant concerns need to be addressed.

Major concerns:

1. The expression pattern of SCNA in mouse and human kidney should be described in greater detail. Of note, the Human Protein Atlas shows discrete staining of podocytes and no staining of tubular cells with two different SCNA antibodies. The data used to argue that a loss of SCNA expression is associated with renal fibrosis in human kidney disease is totally inadequate. The patients need to be characterised, SCNA and for SCNA and collagen/a-SMA staining quantified, and double staining or serial section staining much better illustrated.

We thank the Reviewer for valuable comments and suggestions that we think significantly improved the quality of the manuscript.

Response: We have now described in more detail the expression pattern of SNCA in mouse and human kidneys. We have also added additional photomicrographs illustrating

the presence of SNCA in the mouse kidney (**Supplementary Fig. 4**). It is of interest to mention that in our study we have used 2 different antibodies against SNCA to investigate the expression of SNCA. The following antibodies are: #610786 (BD Biosciences) and #4179S (Cell Signaling).

Regarding the human study part, we agree with the reviewer that the characterization of patients was needed and we have now included requested data in the **Supplementary Table 1**. We have also quantified the immunoperoxidase staining's for SNCA, α SMA, as well as Sirius red staining and we have presented the results in the **Figure 5B, C, D**. We have also invested additional efforts and repeated stainings for SNCA, α -SMA and FSP1 in order to make better representative photomicrographs of serial section staining's and more clearly illustrate our results (**Figure 5A**).

2. While the *in vitro* studies argue that loss of SCNA expression leads to tubular cell de-differentiation towards a mesenchymal phenotype, this is not analysed *in vivo*. Do the SCNA negative tubular cells in areas of fibrosis show *de novo* expression of vimentin or Snail1 and a loss of E-cadherin and cytokeratin? This has not been investigated in either the human or mouse studies.

Response: We thank the Reviewer for highlighting this important point. We have performed the experiments of triple immunofluorescence staining for SNCA, E-cadherin and vimentin in kidney sections of contralateral non-obstructed (control) and obstructed (UUO) kidneys (15 days of UUO). The results are presented in the **Supplementary Fig. 5A-D**. We were able to detect certain tubular cells in the area of fibrosis that lost SNCA expression and gained *de novo* vimentin expression, still preserving some levels of E-cadherin expression. Our results *in vivo* confirm a newly established concept of partial EMT program and its contribution to the development of renal fibrosis.

3. The efficiency of SCNA deletion in PEPCK expressing tubules has not been quantified.

Response: We agree with the Reviewer that this is important information. We have now quantified the efficiency of SNCA deletion in PEPCK expressing tubules and included it in the **Figure 6B**.

4. (A) The conclusion that loss of RPTEC expression of SCNA leads to greater renal fibrosis in the UUO model is not certain. Fig 6G shows no difference in Sirius red staining area in the day 5 obstructed kidney between the two genotypes. Fig 6H shows a 2-fold increase in collagen deposition in the contralateral control kidney on day 15 of UUO in the KO group – which was not evident at day 5 UUO. What has happened here? Do these mice spontaneously development renal fibrosis with age? Also, this means that the fold difference in Sirius red staining between the obstructed and non-obstructed kidney in the KO mice (approx. 2.5 fold) is less than that seen in the WT control mice (approx. 4-fold).

Response: To respond to the question whether Cre⁺SNCA Flox/Flox mice spontaneously develop fibrosis with age, we analysed the expression of key fibrosis markers at the mRNA and protein level in the kidneys of Cre⁺SNCA Wt/Wt and Cre⁺SNCA Flox/Flox mice at the

age of 15 weeks (**R Fig. 2**, figure added at the end of this file). Given the time necessary to generate these animals and the fact that at the time we received the response from the Editor regarding the review of our manuscript we were experiencing difficulties producing sufficient number of animals for the experiment, we only had mice at the age of 15 weeks (as the oldest ones) to perform this experiment. We have detected a tendency toward increase of certain fibrosis markers at the level of mRNA in Flox mice compared with Wt mice, which was not statistically significant. We have also detected a tendency toward increase of α SMA and vimentin protein expression in the kidneys of Flox mice compared with Wt counterparts. At this point, we cannot say that Cre⁺SNCA Flox/Flox mice spontaneously develop renal fibrosis with age. To investigate whether they develop fibrosis with age would require more profound analysis that is beyond the scope of this paper. Nevertheless, it is known that unilateral obstruction can affect the non-obstructed kidney (Hauser et al. 2005; PMID: 16316326). Obstructed kidney may release messengers into the blood stream causing contralateral activation of different genes. So, in our UUO model, mice that do not have SNCA in the proximal tubule not only show more aggravated renal fibrosis in the obstructed kidney, but also show more exaggerated transcriptional and translational response in the contralateral kidney (Fig. 5 and 6 – previous version of the manuscript and Fig. 6 and Fig.7 revised version of the manuscript). The longer UUO lasts, the more visible changes are seen. So, after 5 days UUO we can see the tendency for fibrosis markers (mRNA) to increase, while at UUO15 this difference becomes statistically significant due to a longer duration of the impact on the contralateral kidney. It seems that the elimination/absence of SNCA by itself is sufficient to make kidney more susceptible to renal fibrosis, as it is suggested by the *in vitro* experiments in which the elimination of SNCA increases the expression of fibrotic markers. This fact, together with the evidences that in the UUO model the contralateral kidney is subjected to profibrotic stimulus (both hemodynamic and humoral), could explain the increase of fibrotic markers in the contralateral kidney of the floxed mice, compared to the contralateral kidney of the control mice. Then the levels of expression of fibrotic markers would be submaximal with both interventions (elimination of SNCA and obstruction) and thus, the difference with the contralateral non-obstructed kidney seems to be smaller than in the WT mice in which the non-obstructed kidney still preserves the expression of SNCA. In any case, the new *in vivo* model confirms that animals with elimination of SNCA in the RPTEC show greater renal fibrosis than control mice.

Regarding Reviewer's comment on difference in Sirius red staining mentioned above, to confirm our findings, we have repeated quantifications for Sirius Red staining's (and obtained the same results) and we have also performed additional staining for collagen I in mouse kidney slices which we explain in more details below in the following comment.

(B) It would be helpful to show the actual area stained rather than fold change. In addition, analysis of deposition of a specific type of collagen would be helpful to validate the Sirius red staining.

Response: We have now presented results for Sirius Red staining as an actual area stained for both time points (UUO5 and UUO15) (**Figure 7K, L**). We have also done the same for any other Sirius red quantification results in the manuscript. To validate the Sirius red staining, we have also performed the immunoperoxidase staining for Collagen I in the kidney slices from Cre⁺ SNCA Wt/Wt and Cre⁺ SNCA Flox/Flox at both time points after

the UUO. The results were quantified and presented as a % of stained area for Collagen I vs. total area (**Figure 7M, N**). The representative photomicrographs of immunoperoxidase staining for Collagen I were also presented in the new **Figure 7O, P**. Results obtained after immunoperoxidase study corroborate the results obtained after Sirius Red staining in UUO model of renal fibrosis in Cre⁺ SNCA Wt/Wt and Cre⁺ SNCA Flox/Flox mice.

(C) Most importantly, analysis of a second model of renal interstitial fibrosis is required to validate this possible finding.

Response: We thank the Reviewer for bringing this critical point to our attention. To validate our findings, we extended our studies to an additional model of renal fibrosis, adenine-induced tubulointerstitial nephropathy in mice. In the **Supplementary Fig. 7A-M**, we show that RPTEC-specific deletion of SNCA aggravates adenine-induced renal fibrosis. The results from the second model of renal fibrosis confirmed the results obtained from the UUO model. Using two mechanistically distinct mouse models of renal fibrosis we confirm that SNCA from RPTECs has a role in protecting kidney parenchyma against fibrosis.

5. Fig 5 – the Western blots for collagen I shown in Fig 5A & B do not concord with the graphs of the blots shown in Fig 5C & E. The blots indicate a substantial increase in collagen I content in the control kidney of mice with SCNA conditional deletion compared to the control kidney of non-deleted mice – but there is no difference in the blots. This does not correlate with the Sirius red staining data. Also, a 70kDa band is rather small for collagen I.

Response: When commenting western blots for collagen I in the comment 5, we believe reviewer #2 refers to Figure 6A, B, C & E presented in the initial version of the manuscript (Fig. 5 describes other results). In this context, we have revised our previous western blots for Collagen I and we have repeated all western blot testing's for collagen I expression at both time points (UUO5 & UUO15). We have also done densitometry for the newly performed western blots. We have also repeated quantifications for Sirius red staining of mouse kidney sections to additionally confirm our findings. In the Fig. 6A (UUO5) and Figure 6B (UUO15) (previous version of the manuscript), presented blots indeed show an increase in Collagen I content in the contralateral kidney of SNCA F/F mice, nevertheless after normalizing for GAPDH in densitometric analysis of the blots, the difference disappeared or it is not statistically significant (Fig. 6 C, E). After repeating western analysis for UUO5 and UUO15 we have obtained technically much better blots and we presented new representative blots in the revised version of the manuscript (**Figure 7A, B**). Our western blot data for the expression of Collagen I at the time point UUO5 agree with Sirius red staining data (previous Fig. 6G), as we did not find statistical difference in the Collagen I content in CTL kidneys of the two investigated groups of mice. After performing new western blots and densitometry analysis for the time point UUO5, we confirm the results we had before (**Figure 7A, F** revised manuscript) and the result agree with Sirius red staining (**Figure 7K**, revised manuscript). Regarding the results for collagen I expression for the time point UUO15 presented in the Figure 6B, E (initial version of the manuscript), we agree with the reviewer that our western blot data for the expression of Collagen I after UUO15 in CTL kidneys of Flox mice do not match with Sirius red staining

(Fig. 6H, initial version). After repeating western analysis for UUO15 and densitometry of newly performed western blots, our western blot data for collagen I agree with Sirius red staining data regarding the collagen I content in the control kidney of mice with SNCA conditional deletion. As we mentioned before, we have also performed immunoperoxidase staining for Collagen I in the kidney slices from Cre⁺ SNCAWt/Wt and Cre⁺ SNCAFlox/Flox at both time points and corroborated one more time the results regarding the collagen accumulation in the kidney, as the results from western blot for collagen I, Sirius red and immunoperoxidase staining for collagen I fully concord between each other's in all groups of mice. Regarding the molecular size of Collagen I, in our study we used the antibody AB765P (Chemicon/Merck Millipore). According to the antibodies' datasheet, the MW of mature Collagen Type I is 70-90 kDa. Olujo *et al.* 2014 (PMID: 25313562) using the same antibody detected the band for Collagen I around 90 kDa in the kidney after UUO. In our blots we are detecting the band above 70 kDa, which was accidentally mislabeled in the representative blots for collagen I (Fig. 6A and B, initial version). We apologize for this mistake and we have now corrected it in the revised version of the manuscript (**Figure 7A, B**, revised version of the manuscript).

6. In Fig 3, why should knock-down of SNCA affect TGF- β 1 induced p38 or Akt signalling? How does this operate?

Response: We thank both Reviewers (See also Reviewer #3, Comment: 5) for bringing this critical point to our attention. We have now performed additional experiments to assess the possible mechanism that stands behind the enhanced activation of p38 and Akt after SNCA knockdown. The activation of MMK3 and MMK6, two closely related dual-specificity protein kinases known to phosphorylate p38 MAP kinase, did not change significantly after knockdown of SNCA in HK-2 cells, so an increase in activation of MKK3/6 as the responsible for the increase of phosphorylation of p38 is unlikely (**Supplementary Fig. 3B**). Next, we performed colocalization and co-immunoprecipitation experiments to investigate possible interactions between SNCA and p38 MAPK. As shown in **Figure 3F**, p38 MAPK colocalized with SNCA in HK-2 cells in basal conditions. Moreover, co-immunoprecipitation experiments using cell lysates from HEK293T cells co-transfected with SNCA-flag and p38-HA confirmed the binding of SNCA with p38 MAPK (**Figure 3G**). As we commented in the Discussion section of the manuscript, based on the fact that SNCA binds to p38 MAPK, it is likely that the SNCA affects phosphorylation of p38 acting as a scaffold protein, inhibiting phosphorylation when bound. The binding of SNCA to p38 could explain not only the activation of p38 upon SNCA downregulation, but also the subsequently activation of Akt in HK-2 cells. Namely, it has been shown that p38 pathway can influence Akt signaling, placing Akt kinase downstream of p38 MAPK^{1, 2}. Cabane *et al.*² demonstrated that stimulation of p38 resulted in a simultaneous activation of Akt pathway, while the inhibition of p38 led to a reduction of pAkt levels. Interestingly, neither inhibition nor activation of Akt had any effect on p-p38 levels in C2C12 cells². McGuire *et al.*¹ confirmed the results from previous group demonstrating the existence of a cross talk between the p38 α -MK2/3 and Akt pathways in macrophages and described a novel role for p38 α -MK2/3 in regulating TLR-induced Akt activation in macrophages. The results obtained here depict the role for SNCA in regulating MAPK-p38 and PI3K-Akt pathways and are in line with previously published data in CNS³⁻⁵. Namely, Iwata *et al.* 2011³ demonstrated that SNCA affects the phosphorylation of MAPKs and that binding of

SNCA to p38 MAPK regulates MAPK pathway by reducing the amount of available p38 to be phosphorylated. Furthermore, Musgrove *et al.*⁵ showed that SNCA prevents cytochrome c release and apoptosis through inhibition of the MAPK signaling pathway.

Does over-expression of SNCA also affect these pathways?

We have now assessed the expression of signalling molecules (pAkt, pErk and p-p38) in HK-2 cells overexpressing SNCA. The results show that overexpression of SNCA did not significantly affect the activity of these pathways at basal state (**Supplementary Fig. 3A**). However, there was a clear tendency to decrease the activation of p38 and Akt after the cells were stimulated with TGF- β 1 and an inhibition of the TGF- β 1-induced expression of fibrotic markers (**Figure 2F**).

Reviewer #3 (Remarks to the Author):

This manuscript described an interesting role for Alpha-(α)-synuclein (SNCA) in protecting against profibrotic changes of proximal tubular epithelial cells via inhibition of MAPK-p38 and PI3K-Akt pathways in EMT and ECM accumulation. It is well designed study with both in vitro and in vivo evidence to support their conclusions.

We thank the Reviewer for the positive assessment of our work.

1. Morphological changes of HK-2 cells as evidence of TGF= β 1-induced EMT is required.

Response: We have now included photomicrographs of HK-2 cells upon treatments with different concentrations of TGF- β 1 and we mentioned the result in the Result section. As seen from the **Supplementary Fig. 2A, B**, TGF- β 1 induced visible changes in cell morphology toward the loss of cobble-stoned shape and acquisition of spindle-like form of loosely interconnected cells.

2. It is not clear what time points were samples analysed for E-cadherin, Vimentin, α -SMA in figure 1B. In time sequence of EMT in the in vitro study, is downregulation of the SNCA an early or late event.

Response: Figure 1B analyses the protein expression of E-cadherin, Vimentin, α -SMA and SNCA after 72 hours of incubation with TGF- β 1. In the Figure 1A, we show different expression levels of SNCA mRNA after treatment with various concentrations of TGF- β 1 during 24, 48 and 72 hours. Analysing different concentrations of TGF- β 1 at different time points, we detected the first decrease of SNCA mRNA 48 h after the beginning of the TGF- β 1 treatment which could be considered as a later event (or later intermediate event) in the time course of EMT.

3. Fig 2A does show a TGF= β 1-induced increase of α -SMA in shSNCA infested HK2 cells.

The blots show a tendency toward increasing α -SMA expression in cells carrying SNCA shRNA upon treatment with TGF- β 1. However, after evaluation of several blots and the normalization with tubulin band, that increase was not statistically significant. Thus, it seems that the increase in α -SMA induced by the deletion of SNCA is already submaximal in the conditions used.

4. Fig 2B overexpression of SNCA showed a clear non-responsiveness of infested HK2 cells in TGF-beta1-induced increase of α -SMA and Vimentin. Would this result suggest a different mechanism to that of SNCA deficiency by shSNCA?

Response: We believe that the mechanism is the same. Thus, overexpression of SNCA will block the induction of EMT by TGF- β 1 in the same way that elimination of SNCA will induce the EMT independently of the stimulus (TGF- β 1). Thus it seems that SNCA is in the EMT signaling pathway, downstream of TGF- β 1.

5. The study demonstrated for the first time that TGF-beta down regulate SNCA via MAPK-p38 activation. What is the mechanism for the demonstrated SNCA inhibition of Akt and p38 activation which are both downstream of TGF-beta as MPAK-p38.

Response: We thank both Reviewers (See also Reviewer #2, Comment: 6) for bringing this critical point to our attention. We have now performed additional experiments to assess the possible mechanism that stands behind the enhanced activation of p38 and Akt after SNCA knockdown. The activation of MMK3 and MMK6, two closely related dual-specificity protein kinases known to phosphorylate p38 MAP kinase, did not change significantly after knockdown of SNCA in HK-2 cells, so an increase in activation of MKK3/6 as the responsible for the increase of phosphorylation of p38 is unlikely (**Supplementary Fig. 3B**). Next, we performed colocalization and co-immunoprecipitation experiments to investigate possible interactions between SNCA and p38 MAPK. As shown in **Figure 3F**, p38 MAPK colocalized with SNCA in HK-2 cells in basal conditions. Moreover, co-immunoprecipitation experiments using cell lysates from HEK293T cells co-transfected with SNCA-flag and p38-HA confirmed the binding of SNCA with p38 MAPK (**Figure 3G**). As we commented in the Discussion section of the manuscript, based on the fact that SNCA binds to p38 MAPK, it is likely that the SNCA affects phosphorylation of p38 acting as a scaffold protein, inhibiting phosphorylation when bound. The binding of SNCA to p38 could explain not only the activation of p38 upon SNCA downregulation, but also the subsequent activation of Akt in HK-2 cells. Namely, it has been shown that p38 pathway can influence Akt signaling, placing Akt kinase downstream of p38 MAPK^{1, 2}. Cabane *et al.*² demonstrated that stimulation of p38 resulted in a simultaneous activation of Akt pathway, while the inhibition of p38 led to a reduction of pAkt levels. Interestingly, neither inhibition nor activation of Akt had any effect on p-p38 levels in C2C12 cells². McGuire *et al.*¹ confirmed the results from previous group demonstrating the existence of a cross talk between the p38 α -MK2/3 and Akt pathways in macrophages and described a novel role for p38 α -MK2/3 in regulating TLR-induced Akt activation in macrophages. The results obtained here depict the role for SNCA in regulating MAPK-p38 and PI3K-Akt pathways and are in line with previously published data in CNS³⁻⁵. Namely, Iwata *et al.* 2011³ demonstrated that SNCA affects the phosphorylation of MAPKs and that binding of

SNCA to p38 MAPK regulates MAPK pathway by reducing the amount of available p38 to be phosphorylated. Furthermore, Musgrove *et al.*⁵ showed that SNCA prevents cytochrome c release and apoptosis through inhibition of the MAPK signaling pathway.

6. *in vitro* study using HK2 cells may not support *in vivo* activation of myofibroblasts, as contribution of tubular epithelial cells towards myofibroblasts has been questioned by other researchers.

Response: Despite the widely accepted role of TGF- β 1 in initiating epithelial-mesenchymal transition (EMT) *in vitro*, and the fact that initial cell-fate tracing experiments described significant contribution of renal epithelial cells to myofibroblasts *in vivo* (PMID: 12163453), subsequent studies indeed questioned the role of EMT in fibrogenesis *in vivo* (PMID: 20008127; 20150430), therefore questioning the sole existence of the EMT process. Nevertheless, recent studies (using lineage tracing, co-staining, and flow cytometry analysis) confirmed that tubular epithelial cells (TECs) do undergo EMT process *in vivo*, but they remain inside the tubule, displaying a partial EMT status (PMID: 26236989; 26236991). Namely, TECs affected by partial EMT express mesenchymal markers while still preserving the expression of epithelia-related genes. Acquiring partial EMT features dramatically affects functionality of TECs, change their protein expression and secretome profile. TECs undergoing this phenotypic switch attract inflammatory cells and activate fibroblasts and pericytes in a paracrine manner, making tubular epithelial cells an important mediator of renal fibrosis, rather than a "bystander" in the process (PMID: 30425237).

References

1. McGuire, VA, Gray, A, Monk, CE, Santos, SG, Lee, K, Aubareda, A, Crowe, J, Ronkina, N, Schwermann, J, Batty, IH, Leslie, NR, Dean, JL, O'Keefe, SJ, Boothby, M, Gaestel, M, Arthur, JS: Cross talk between the Akt and p38 α pathways in macrophages downstream of Toll-like receptor signaling. *Mol Cell Biol*, 33: 4152-4165, 2013.
2. Cabane, C, Coldefy, AS, Yeow, K, Dérijard, B: The p38 pathway regulates Akt both at the protein and transcriptional activation levels during myogenesis. *Cell Signal*, 16: 1405-1415, 2004.
3. Iwata, A, Maruyama, M, Kanazawa, I, Nukina, N: alpha-Synuclein affects the MAPK pathway and accelerates cell death. *J Biol Chem*, 276: 45320-45329, 2001.
4. Seo, JH, Rah, JC, Choi, SH, Shin, JK, Min, K, Kim, HS, Park, CH, Kim, S, Kim, EM, Lee, SH, Lee, S, Suh, SW, Suh, YH: Alpha-synuclein regulates neuronal survival via Bcl-2 family expression and PI3/Akt kinase pathway. *FASEB J*, 16: 1826-1828, 2002.
5. Musgrove, RE, King, AE, Dickson, TC: α -Synuclein protects neurons from apoptosis downstream of free-radical production through modulation of the MAPK signalling pathway. *Neurotox Res*, 23: 358-369, 2013.

R Fig. 1

R Fig. 1. Transmission electron microscopy (TEM) analysis of kidney sections from wild-type mice showing subcellular distribution of SNCA. Three magnifications are shown for each sample (A, D) $\times 3k$, (B, E) $\times 20k$, (C, F) $\times 80k$. Red squares represent the areas of interest with the positive SNCA staining. PCT, proximal convoluted tubule; DCT, distal convoluted tubule; N, nucleus; M, mitochondria; Mv, microvilli on the brush border of the proximal tubule cell, BM, basement membrane; BC, Bowman's capsule; P, podocyte; Pc, podocyte pedicels.

R Fig. 2

R Fig. 2. Expression of fibrotic markers in PEPCK^{Cre+} SNCA^{wt/wt} and PEPCK^{Cre+} SNCA^{fl/fl} mice with age. (A) Total mRNA was extracted from kidneys of PEPCK^{Cre+} SNCA^{wt/wt} and PEPCK^{Cre+} SNCA^{fl/fl} mice at the age of 15 weeks. mRNA levels of collagen I, fibronectin, αSMA, vimentin, TGF-β1 and cadherin-16 were determined by quantitative real-time PCR and normalized to TBP. Data are presented as mean ± SEM (3-4 mice/group). (B, C) Whole kidney lysates from kidneys were processed for protein analysis and were immunoblotted with antibodies against αSMA, vimentin, and GAPDH. (B) Representative Western blot analysis of αSMA and vimentin in the kidneys of PEPCK^{Cre+} SNCA^{wt/wt} and PEPCK^{Cre+} SNCA^{fl/fl} mice at the age of 15 weeks. (C) Quantitative analysis by densitometry. Data were normalized to GAPDH and presented as mean ± SEM (3-5 mice/group). (D) Representative images of immunoperoxidase staining for αSMA, FSP1 and Sirius Red staining in mouse kidney sections. Original magnification x20. (E) Quantification of collagen content after Sirius red staining was expressed as a positive stained area vs. total analyzed area. Data are presented as mean ± SEM (3-5 mice/group).

Reviewers' comments:

Reviewer #2 (Remarks to the Author):

The manuscript has been improved, but I still have major concerns regarding evidence of tubular de-differentiation, the human data, and the proposed mechanism of tubular de-differentiation.

Comments:

1. The SNCA staining of tubular cells in the new Fig 5A is not convincing. In the fibrotic kidney, there are numerous interstitial SNCA+ cells; this is a new finding and opens the question of what cell types are they? The scoring of SNCA expression is based on a basic mild/weak/strong assessment of SNCA immunostaining; however, it is not indicated whether this scoring relates to the whole tissue section or only to tubular cells, or does this scoring include the interstitial cells? Indeed, the histoscore appears insensitive when looking at the α -SMA staining. At a bare minimum, there should be a correlation analysis of the percentage of SNCA+ tubules and the degree of α -SMA staining and Sirius red staining. Is there de novo vimentin expression by SNCA negative tubules? Does the extent of tubular SNCA staining correlate with renal function? The Fig 5 legend indicates that only 30 patient biopsies were assessed according; however, 52 patients are listed in Table 1 – which patients were not used, or were different patients used for different antibody staining? Table 1 is incomplete: what are “other KD” patients, the eGFR (and proteinuria data) should be provided for each patient group.
2. Suppl Fig 4 shows one tubular cell which lacks SNCA staining and has clear vimentin staining. However, most tubular cells that lack SNCA do not appear to express vimentin. Showing a single cell is not a convincing argument that tubules lacking SNCA undergo de-differentiation in the UO model.
3. Characterisation of SNCA deletion in PEPCK expressing tubular cells is much improved.
4. The collagen I immunostaining provided in the new Fig 7 is of poor quality. Collagen I is not detected in normal kidney, while better quality images are needed to demonstrate that the collagen I staining in the UO kidney is actually extracellular.
5. The second model of renal fibrosis (4-week adenine diet model) strengthens the conclusions of the study. This data should be in the main manuscript rather than in the Suppl figures. What happened to renal function in this model, and was there evidence of partial EMT? Is there any glomerular damage in this model?
6. The AB765P antibody detects 2 bands of collagen I of >130kDa (see Kidney International (2015) 88, 515–527). Recombinant forms of collagen I can be 70kDa, but native collagen I prepared from tissue lysates is significantly larger.

7. Supplementary Fig. 3B – this shows the level of phosphorylation of MKK3/6. However, this is not the same as demonstrating activity of MKK3/6. It is entirely possible that the increase in p-p38 seen with SNCA knockdown is due to MKK3/6; this cannot be discounted. Indeed, what other enzymes would be responsible for p38 phosphorylation? The argument is that SNCA sequesters p38 to limit its activation by MKK3/6. Was p38 phosphorylation evident in tubular cells lacking SNCA, but lacking in SNCA positive cells in either human or mouse studies?

Reviewer #3 (Remarks to the Author):

I have no further concerns.

Reviewer #4 (Remarks to the Author):

The manuscript is describing potential protective role of alpha-synuclein in the pathogenesis of renal fibrosis. The manuscript is potentially interesting and manuscript is improved by revision. However, some key points were not addressed. Thus:

1. The lower expression level of gamma and beta-synuclein in these cells does not mean that it is not functional and function of these synucleins could lead to pathology. Aggregated synuclein (oligomeric) shown to be toxic in picomolar concentration. Considering this effect of synuclein (positive or negative) can be assessed only in triple synuclein deficiency
2. In continuing of the above comment - we cannot exclude toxicity of oligomeric synucleins (beta and gamma) in alpha-syn deficiency - authors should still comment on this and better measure with specific antibodies.
3. The authors focused on the MAPK-p38 and PI3K-Akt pathways and synuclein. However, alpha-synuclein shown to have a physiological role (in monomeric state) in neuronal tissue which is not always reflect only specific neuronal function - thus, role in cell signaling and mitochondrial bioenergetics was demonstrated. Authors should comment if it can be (or not) in retinal cells

Reviewer #2 (Remarks to the Author):

The manuscript has been improved, but I still have major concerns regarding evidence of tubular de-differentiation, the human data, and the proposed mechanism of tubular de-differentiation.

Response: We thank the Reviewer for positive evaluation of our work and effort.

Comments:

1. (A) The SNCA staining of tubular cells in the new Fig 5A is not convincing. In the fibrotic kidney, there are numerous interstitial SNCA+ cells; this is a new finding and opens the question of what cell types are they?

Response: We agree with the reviewer that on the photomicrograph of the fibrotic kidney presented in the Figure 5A we can see some interstitial cells in the kidney stained for SNCA. Indeed, it has already been reported that SNCA is expressed by a variety of peripheral tissues and cells such as lymphocytes, monocytes (Shin *et al.* 2000, PMID 10774749; Shameli *et al.* 2016, PMID 26517968) and fibroblast (Hoepken *et al.* 2008, PMID: 18511044). Furthermore, interstitial SNCA+ cells can also be seen in the intertubular space of healthy areas of the human kidney (**R Fig. 5**, attached at the end of this file for the reviewer's evaluation). Therefore, several other cell types of the kidney (apart from RPTECs) can express SNCA and may as well preserve the expression of SNCA after renal fibrosis, as we already mentioned in the Results regarding SNCA expression in glomeruli. Our focus in the present study is to assess the role of SNCA in tubular cells of the kidney and, therefore, we believe that addressing the question of which cell type are cells stained in the interstitial space is beyond the scope of this study. We think that results from our *in vitro* study, as well as the data from our extensive *in vivo* study using two different models of renal fibrosis and a specific mouse model with conditional gene silencing of SNCA in RPTECs, give sufficiently hard evidence on the role of SNCA in renal tubular cells and its involvement in renal fibrosis.

(B) The scoring of SNCA expression is based on a basic mild/weak/strong assessment of SNCA immunostaining; however, it is not indicated whether this scoring relates to the whole tissue section or only to tubular cells, or does this scoring include the interstitial cells? Indeed, the histoscore appears insensitive when looking at the a-SMA staining.

Response: The scoring of SNCA staining presented in the Figure 5B is related to staining of SNCA in tubular cells, and not in interstitial cells. When commenting Figure 5A, B in the main text of the manuscript we mentioned that, and point out that SNCA decreases in the dilated renal tubules. Furthermore, in the Figure Legend of the Figure 5 we have now cleared out that quantification of SNCA staining refers to staining of renal tubular cells.

(C) At a bare minimum, there should be a correlation analysis of the percentage of SNCA+ tubules and the degree of α -SMA staining and Sirius red staining.

Response: We agree with the reviewer and we have now performed the correlation analysis between the percentage of SNCA+ tubules and the degree of α SMA and Sirius red staining. We have found a significant correlation between the percentage of SNCA+ tubules and α SMA staining in human kidney tissue samples ($p=0.0043$; $R= - 0.453$), as well as the significant correlation between the percentage of SNCA+ tubules and Sirius red staining ($p=0.0061$; $R= - 0.462$). The results are presented in the **Figure 5E and F**, respectively.

(D) Is there *de novo* vimentin expression by SNCA negative tubules?

Response: We have now performed the experiments of immunoperoxidase staining for SNCA and vimentin in serial sections of human kidneys. The results are presented in the **Supplementary Fig. 6**. We were able to find dilated tubules in the areas of fibrosis with tubular cells showing no expression for SNCA while presenting *de novo* expression of vimentin. Besides Supplementary Fig. 6, we have attached some additional images of immunoperoxidase staining for SNCA and vimentin in the **R Fig. 1** (at the end of this file) for the reviewer's evaluation.

(E) Does the extent of tubular SNCA staining correlate with renal function?

Response: We have now performed the correlation analysis between the extent of tubular SNCA staining and renal function (GFR, mL/min/1.73 m²) and we have not found significant correlation between these two parameters. The results are attached for reviewer's evaluation at the end of this file in the **R Fig. 4**. The results however are not surprising as the degree of renal dysfunction is variable depending of the etiology of renal failure and were only meant to confirm in human samples that SNCA expression was decreased in altered tubular cells.

(F) The Fig 5 legend indicates that only 30 patient biopsies were assessed according; however, 52 patients are listed in Table 1 – which patients were not used, or were different patients used for different antibody staining?

Response: Thank you very much for pointing out this error. When we started our human study we obtained patient data and characteristics for 52 individuals. We initially performed staining's in 30 patients because for 30 patients we have sufficient number of slides for all staining we wished to performed. The number of patients 52 stayed by error in the Table 1. We have now obtained more human kidney tissue slides from Biobank (from the initially characterized 52 patients) and we have performed additional staining and included them in our human study. We would like to emphasize that every patient was analysed for all staining's (SNCA, α SMA, FSP1, Sirius Red). We have now revised Table 1 and excluded patients whose samples were not used for staining analysis.

(G) Table 1 is incomplete: what are “other KD” patients, the eGFR (and proteinuria data) should be provided for each patient group.

Response: The term “Other KD” patients in the Table 1 encompasses patients with focal segmental glomerulosclerosis (n=2), diabetic nephropathy (n=2), membranous nephropathy (n=2) and minimal change disease (n=1). Due to a small number of patients in each group we decided not to put them as separate entities in the Table 1, but as one group called “other KDs”. We have now explained the meaning of “other KDs” in the new Table 1. We have also introduced eGFR and proteinuria data for each patient group in the new Table 1, as suggested by the reviewer.

2. Suppl Fig 4 shows one tubular cell which lacks SNCA staining and has clear vimentin staining. However, most tubular cells that lack SNCA do not appear to express vimentin. Showing a single cell is not a convincing argument that tubules lacking SNCA undergo de-differentiation in the UUO model.

Response: When commenting staining for SNCA and vimentin in the comment 2, we believe reviewer #2 refers to Supplementary Fig. 5 (not Suppl Fig 4, as mentioned in the comment 2). Our intention with Supplementary Fig. 5 was to show on the same image the presence of one cell that lost SNCA expression and gained *de novo* expression of vimentin, but at the same time the presence of cells that still express some levels of SNCA and do not have expression of vimentin. It was a very difficult task and we managed to illustrate it in the Supplementary Fig. 5, nevertheless, we obviously did not explain it well. The presence of tubular cells that preserve SNCA staining but do not express vimentin is best illustrated in Supplementary Fig.5D (where we have SNCA/Vimentin staining presented) rather than in the Supplementary Fig. 5C (where we have SNCA/ECdh staining presented) due to an overlapping of SNCA and E-cadherin stainings on the panel C. We indeed pointed here to one cell that lost SNCA expression and gained vimentin expression, but showing one cell does not mean this is the only cell we found in all our staining’s. Besides Supplementary Fig. 5 that we left in the Supplementary Material, we now attached some additional images of triple immunofluorescence staining in the **R Fig. 2** (at the end of this file) for the reviewer’s evaluation. In any case, we would like to remind that epithelial-mesenchymal transition is a highly dynamic process. Thus, during this process cells are losing the epithelial phenotype while gaining mesenchymal properties and therefore, sequentially activating and deactivating gene expression. Thus, it is possible to find cells that have lost some epithelial markers and still have not started expressing some mesenchymal ones. It is quite a task to catch some transitional moments even in *in vitro* settings where the conditions are much more controlled, let alone in *in vivo* milieu where so many variables influence a single event. In our immunofluorescence staining’s we managed to detect tubular epithelial cells in dilated kidney tubules with decreased or even lost expression of SNCA and *de novo* expression of vimentin. However, more detailed *in vivo* experiments

are needed to prove that tubular epithelial cells displaying a partial EMT status lose SNCA and gain vimentin, and they represent a huge task that would require lineage-tracing experiments that are beyond the scope of this paper.

Notwithstanding, we would again like to thank the reviewer #2 for suggesting us to perform these triple immunofluorescence experiment *in vivo* for the reason that now we managed to find evidence in 3 different models (UUO-induced renal fibrosis, adenine diet-induced fibrosis and human kidney fibrosis) that tubular cells from dilated renal tubules in the areas of fibrosis not only are losing expression of SNCA (which we already demonstrate here), but are also able to express *de novo* mesenchymal marker vimentin.

3. Characterisation of SNCA deletion in PEPCK expressing tubular cells is much improved.

Response: Thank you.

4. The collagen I immunostaining provided in the new Fig 7 is of poor quality. Collagen I is not detected in normal kidney, while better quality images are needed to demonstrate that the collagen I staining in the UUO kidney is actually extracellular.

Response: We have analysed again our immunoperoxidase staining for collagen I in kidney sections of UUO kidneys and corresponding controls and made new, higher quality images to better illustrate the collagen I staining. Previous images of collagen I (Figure 7, revision 1) have now been replaced with new images in **Figure 7O** (UUO5) and **Figure 7P** (UUO15). In order to corroborate our findings regarding collagen I accumulation in kidneys of SNCA Wt and Flox animals, we have now performed immunoperoxidase staining and quantification for Collagen I in kidney sections of our adenine model of renal fibrosis, which we did not have before. In that way, we have complemented our study with this additional piece of information. These results are included in **Figure 8L, N**.

5. (A) The second model of renal fibrosis (4-week adenine diet model) strengthens the conclusions of the study. This data should be in the main manuscript rather than in the Suppl figures.

Response: We acknowledge the reviewer's suggestion and we have now presented the results from the second model of renal fibrosis (4-week adenine diet model) in the main manuscript (**Figure 8**).

(B) What happened to renal function in this model, and was there evidence of partial EMT?

Response: Regarding the renal function in this model, we measured levels of BUN in the serum of all animals after 2, 3 and 4 weeks of adenine diet feeding. We have found statistically significant increase of serum BUN in both Cre⁺ SNCAWt/Wt and Cre⁺

SNCAFI/Fl mice fed an adenine diet over a period of 4 weeks, compared with their counterparts fed a regular diet. There was no significant difference in levels of serum BUN between Cre⁺ SNCAWt/Wt and Cre⁺ SNCAFI/Fl mice fed an adenine diet. The results are presented in the **Supplementary Fig. 8**.

Regarding partial EMT in the adenine model of renal fibrosis, we have now performed experiments of triple immunofluorescence staining for SNCA, E-cadherin and vimentin in kidney sections of Cre⁺ SNCAWt/Wt mice fed a regular or adenine diet. The results are presented in the **Supplementary Fig. 9 (A-D)**. Furthermore, we attached additional images of triple immunofluorescence staining for SNCA, E-cadherin and vimentin (not presented in the Supplementary Fig. 9) with different fields captured showing various tubular cells in the areas of fibrosis. These images are attached for reviewer's evaluation at the end of this file in the **R Fig. 3**. In the Supplementary Fig. 9 we can see that adenine-rich diet induced fibrosis led to a decrease of SNCA expression in dilated renal tubules where different tubular cells at the same time showed expression of mesenchymal marker vimentin. Our results obtained in adenine diet-induced model of renal fibrosis are in line with the results obtained after UO-induced fibrosis where we also detected tubular cells in the area of fibrosis showing decreased or even lost expression of SNCA while showing positive immunostaining for vimentin.

(C) Is there any glomerular damage in this model?

Response: Since the main focus of our paper was the damage of tubular structures of the kidney and the investigation of the role of SNCA in the maintenance of the epithelial phenotype of renal tubular cells, we have not analysed the glomerular damage neither structural/pathologic changes of glomerulus upon adenine diet feeding.

6. The AB765P antibody detects 2 bands of collagen I of >130kDa (see *Kidney International* (2015) 88, 515–527). Recombinant forms of collagen I can be 70kDa, but native collagen I prepared from tissue lysates is significantly larger.

Response: We thank the Reviewer for the comment. According to the antibodies' datasheet, antibody AB765P (Chemicon/Merck Millipore) detects two bands of collagen I, the band at 140-210 kDa which represents the Collagen I precursor and the band at 70-90 kDa which represents mature collagen I. Collagen is known to derive from large collagen I precursor (140-210 kDa), a complex protein that is subsequently cleaved to smaller derivatives, which can vary between tissues and during different physiological circumstances. During protein folding and secretion, cleavage of N- and C- peptides of collagen I precursor will produce extracellular mature collagen I with molecular weight between 70–100 kDa depending of tissue type, preparation and assessment conditions, as demonstrated in numerous publications (PMID: 25617052, PMID: 16624289, PMID: 24469459, PMID: 24114659, PMID: 25313562). Collagen I from whom the N- and C-

terminals have been removed is secreted and function as the building block for collagen fibril formation. In our western blots we are detecting the band above 70 kDa which corresponds to mature collagen I. Oujó *et al.*, 2014 (PMID: 25313562) using the same antibody are detecting a band for Collagen I around 90 kDa in the tissue lysates from kidneys after unilateral ureteral obstruction (UUO) – the same experimental model as employed in our study. We are aware that in various manuscripts depending on different experimental conditions and needs investigators are detecting intracellular procollagen products or/and extracellular mature collagen. In our experimental conditions/model we are detecting mature collagen I. If necessary for the publication, we can label in the Figure panel that collagen we detect is mature collagen I.

7. (A) Supplementary Fig. 3B – this shows the level of phosphorylation of MKK3/6. However, this is not the same as demonstrating activity of MKK3/6. It is entirely possible that the increase in p-p38 seen with SNCA knockdown is due to MKK3/6; this cannot be discounted. Indeed, what other enzymes would be responsible for p38 phosphorylation? The argument is that SNCA sequesters p38 to limit its activation by MKK3/6.

Response: We thank the Reviewer for bringing this critical point to our attention. First, we have now corrected in the Result section the word *activity* for *phosphorylation* of MKK3/6. Furthermore, we have performed additional experiments to confirm that phosphorylation of p38 seen after SNCA knockdown is due to the capability of SNCA to sequester p38 limiting its activation by MKK3/6. Namely, we co-transfected cells with the MKK6E plasmid carrying constitutive activation (overexpression) of MKK6 kinase as well as with the SNCA-flag plasmid (overexpression of SNCA), and we subsequently analyzed levels of phosphorylated p38. Even when cells had constitutive activation of MKK6 kinase SNCA managed to decrease the levels of phosphorylated p38 (**Supplementary Fig. 3C, D**), confirming that the effects of SNCA on the MAPK pathway are downstream of activation of MKK3/6. Regarding the other possible enzymes responsible for phosphorylation of p38, no other enzymes have been described in the literature. However, an autophosphorylation capacity of p38 has been described. However, this other pathway to increase p38 phosphorylation will be likely inhibited by sequestering p38 by SNCA, so the mechanism of action of SNCA would stay the same.

(B) Was p38 phosphorylation evident in tubular cells lacking SNCA, but lacking in SNCA positive cells in either human or mouse studies?

Response: We have now performed the experiments of double immunofluorescence staining for SNCA and phosphorylated p38 in kidney sections of contralateral non-obstructed (control) and obstructed (UUO) kidneys (5 days of UUO). The results are presented in the **Supplementary Fig. 10A-C**. We were able to detect various tubular cells

in the area of fibrosis that lost SNCA expression (or had it decreased) while showing an increase of p-p38 expression. Healthy SNCA positive tubules showed no staining for phosphorylated p38.

Reviewer #3 (Remarks to the Author):

I have no further concerns.

Response: Thank you.

Reviewer #4 (Remarks to the Author):

The manuscript is discussing potential protective role of alpha-synuclein in the pathogenesis of renal fibrosis. The manuscript is potentially interesting and manuscript is improved by revision.

Response: We thank the reviewer for positive evaluation of our work and effort.

However, some key points were not addressed.

Thus:

1. The lower expression level of gamma and beta-synuclein in these cells does not mean that it is not functional and function of these synucleins could lead to pathology. Aggregated synuclein (oligomeric) shown to be toxic in picomolar concentration. Considering this effect of synuclein (positive or negative) can be assessed only in triple synuclein deficiency.

Response: We appreciate the reviewer's comment and understand the concerns raised here. Taking into consideration the results from the previous revision of the manuscript where we found almost undetectable levels of SNCB and SNCG in HK-2 cells at basal state, the first step we undertook here was to analyze the possible existence of SNCB and SNCG oligomers in the already established experimental settings of SNCA deficiency (see also Reviewer #4, Comment: 2). We analyzed by western blot the presence of oligomeric SNCB and SNCG in HK-2 cells with SNCA deficiency (shSNCA) and we have not detected differences in oligomeric SNCB or SNCG in shSNCA cells compared with control cells (**Supplementary Fig. 1C**). Additionally, we performed immunofluorescence staining's with specific antibodies for SNCB and SNCG and analyzed cells by confocal microscopy analysis. We did not detect an increase or a decrease of SNCB and SNCG staining in HK-2 cells with SNCA deficiency compared with control group of cells (**Supplementary Fig. 1D**). Since the aggregation of proteins is intrinsically concentration dependent (PMID: 28751856) and we were finding almost undetectable protein levels of SNCB and SNCG in HK-2 cells, we decided to additionally assess the mRNA levels of SNCB and SNCG in HK-2 cells with SNCA deficiency. We did not find any significant difference in the levels

of mRNA for both investigated synucleins in HK-2 cells carrying shRNA targeting SNCA compared with control group of cells (**Supplementary Fig. 1E**). Taking into account almost undetectable levels of SNCB and SNCG at basal state, as well as in the settings of SNCA deficiency in HK-2 cells, we feared that undertaking the experiments of triple deficiency would bring us to a technically very difficult task - demonstrating downregulation of SNCB and SNCG in a cell that already express very low or almost undetectable levels of these proteins. We believe that the experiments we performed here could be reliable enough to demonstrate whether upregulation of the other forms of the protein are involved in the effects of eliminating SNCA in HK-2 cells.

2. In continuing of the above comment - we cannot exclude toxicity of oligomeric synucleins (beta and gamma) in alpha-syn deficiency - authors should still comment on this and better measure with specific antibodies.

Response: We thank the Reviewer for this valuable comment. We have now performed additional experiments, as suggested by the reviewer, where we analyzed by western blot the presence of oligomeric SNCB and SNCG in HK-2 cells with SNCA deficiency. We have not found differences in oligomeric SNCB or SNCG in HK-2 cells with SNCA deficiency compared with control cells (**Supplementary Fig. 1C**). Additionally, we performed immunofluorescence staining's with specific antibodies for SNCB and SNCG and analyzed cells by confocal microscopy analysis. We did not detect an increase or a decrease of SNCB and SNCG staining in HK-2 cells with SNCA deficiency compared with control group of cells (**Supplementary Fig. 1D**). Furthermore, we performed qPCR analysis to assess the mRNA levels of SNCB and SNCG in HK-2 cells with SNCA deficiency. We also did not find any significant difference in the levels of mRNA for both investigated synucleins in HK-2 cells carrying shRNA targeting SNCA compared with control group of cells (**Supplementary Fig. 1E**). As suggested by the reviewer, we commented in Discussion part that even though we did not detect differences in the levels of oligomeric synucleins (SNCB and SNCG) in HK-2 cells carrying shRNA for SNCA compared with control ones, we still can not absolutely exclude possible toxicity of these structures in picomolar concentrations. Furthermore, we propose the necessity of further investigation in this direction on an *in vitro* and an *in vivo* level.

3. The authors focused on the MAPK-p38 and PI3K-Akt pathways and synuclein. However, a-synuclein shown to have a physiologicla role (in monomeric state) in neuronal tissue which is not always reflect only specific neuronal function - thus, role in cell signaling and mitochondrial bioenergetics was demonstrated. Authors should comment if it can be (or not) in retinal cells.

Response: We thank the reviewer for this suggestion. It is a very good point and we agree on the possible involvement of SNCA in mitochondrial bioenergetics of renal proximal tubular epithelial cells (RPTECs). Namely, kidney is one of the most energy-demanding

organs in the human body, besides the heart, while RPTECs require more active transport mechanisms than any other renal cell type. Therefore, RPTECs contain more mitochondria than any other structure in the kidney, and the proper maintenance of the mitochondrial homeostasis is critical for the right functioning of the proximal tubule. Numerous studies demonstrate that SNCA is important for mitochondrial homeostasis specifically emphasizing the interplay that exists between mitochondria and SNCA in both physiological and pathological conditions in the central nervous system (PMID: 29292725). Alterations of the protein SNCA, as well as the possible imbalance in SNCA/mitochondria reciprocal-modulatory system can contribute to bioenergetics defects leading to mitochondrial dysfunction and neuronal impairments (PMID: 29292725). Therefore, there is a possibility that SNCA could have a potential role in mitochondrial bioenergetics of RPTECs. We have now appropriately commented on this matter in the Discussion part.

R Fig. 1. Immunoperoxidase staining for SNCA and vimentin in human kidney affected by fibrosis (additional images of fibrotic human tissue areas that complement Supplementary Fig. 6).

R Fig. 2. Triple immunofluorescence staining for SNCA, E-cadherin and vimentin in mouse kidney after UUO-induced renal fibrosis (additional images that complement Supplementary Fig. 5).

R Fig. 3. Triple immunofluorescence staining for SNCA, E-cadherin and vimentin in mouse kidney after adenine-induced renal fibrosis (additional images that complement Supplementary Fig. 9).

R Fig. 4. Correlation analysis between the extent of tubular SNCA staining and renal function (GFR, mL/min/1.73 m2). Linear regression analysis shows no significant correlation between two investigated parameters.

R Fig. 5. Expression of fibrotic markers increases and SNCA decreases in human fibrotic kidney samples. Figure submitted in the first version of the manuscript for NCOMMS.

REVIEWERS' COMMENTS:

Reviewer #2 (Remarks to the Author):

The authors have addressed my concerns.

Comment on the new analysis.

The lack of correlation between SCNA tubular staining and eGFR presumably reflects the rather eclectic group of CKD patients analysed, which contains only a small number of patients with glomerulonephritis. Looking at the data, it appears likely that neither the α -SMA or the picosirius red staining correlate with eGFR in this cohort. Most studies of CKD focus on glomerulonephritis and/or diabetic kidney disease and show a clear correlation between tubulointerstitial fibrosis (α -SMA staining or picosirius red or Masson trichrome staining) and serum creatinine/eGFR. Thus, the authors may have missed an association between the loss of tubular SCNA expression and declining renal function in glomerulonephritis due to the sample set analysed.

Reviewer #4 (Remarks to the Author):

The manuscript is improved by revision and authors addressed all my comments

REVIEWERS' COMMENTS:

Reviewer #2 (Remarks to the Author):

The authors have addressed my concerns.

Response: Thank you for the positive assessment of our work and efforts.

Comment on the new analysis.

The lack of correlation between SCNA tubular staining and eGFR presumably reflects the rather eclectic group of CKD patients analysed, which contains only a small number of patients with glomerulonephritis. Looking at the data, it appears likely that neither the α -SMA or the picrosirius red staining correlate with eGFR in this cohort. Most studies of CKD focus on glomerulonephritis and/or diabetic kidney disease and show a clear correlation between tubulointerstitial fibrosis (α -SMA staining or picrosirius red or Masson trichrome staining) and serum creatinine/eGFR. Thus, the authors may have missed an association between the loss of tubular SCNA expression and declining renal function in glomerulonephritis due to the sample set analysed.

Response: Thank you for your comment. We have now appropriately discussed on this matter in the manuscript text.

Reviewer #4 (Remarks to the Author):

The manuscript is improved by revision and authors addressed all my comments

Response: Thank you for the positive evaluation of our revised manuscript.